# Data driven estimation of gradient fields in the weak sense

## Abstract

Estimating the gradient of a function is crucial in various fields, including data assimilation, reinforcement learning, and generative modeling. These issues are particularly challenging because of the high-dimensional spaces involved. Furthermore, direct gradient computation is often infeasible due to the intrisick lack of the regularity of the underlying function or simply because the function's explicit form is unavailable. In this work, we introduce a novel framework for the offline estimation of gradient fields inspired by score matching. The core idea is to employ an integration-by-parts formulation to replace the direct mean squared error loss with a term involving the divergence of a neural network, which can then be efficiently approximated using Hutchinson's estimator. Numerical experiments include offline gradient estimation where we evaluate the fidelity of estimated gradient for different methods and an application to implicit regression problems involving the two-dimensional Navier–Stokes equations. The results demonstrate the practical potential of the approach for various tasks.

## 1 Introduction

Gradient computation is a fundamental requirement not only in machine learning but also across many scientific disciplines including meteorology, computational fluid dynamics, and control theory. In deep learning, standard techniques such as backpropagation, adjoint methods, and automatic differentiation are routinely used to compute gradients for optimizing complex high-dimensional systems. Similarly, in fields such as weather forecasting and climate modeling, adjoint methods play a crucial role in sensitivity analysis and data assimilation, enabling researchers to infer optimal initial conditions or parameters of the dynamic despite the intrinsic high dimensionality and nonlinearity of these systems.

However, these techniques are not universally applicable. In many practical situations, direct access to accurate gradients is impeded by several factors:

- The function of interest, $J(x)$, may be only accessible as a black box, making classical differentiation infeasible.
- In complex systems such as those governed by stochastic partial differential equations (SPDEs) or large-scale atmospheric models, computational and memory constraints limit the applicability of standard adjoint-based methods.
- Additionally, non-smooth phenomena, such as those found in sea ice dynamics or contact mechanics, can cause classical differentiation methods to fail or yield unreliable gradients.

Lastly, the solutions of geophysical fluid dynamics systems have intrinsically poor regularity; they are only weakly regular. These limitations are particularly critical in contexts where the gradient field itself is the primary object of interest, yet is not directly observable. In **Scientific Machine Learning**, reduced-order models ROMS Lucia et al. (2004) are often employed to emulate complex forecasting dynamics. However, when used for variational inference Artana et al. (2012), the accuracy of the estimated gradient field becomes **crucial**, often outweighing the fidelity of the scalar forecast itself. In **Probabilistic Modeling**,

interpreting the objective function $J(x)$ as an energy potential links the gradient field to the score of a Gibbs measure $p(x) \propto e^{-J(x)}$. Consequently, estimating $\nabla J$ effectively defines a sampler for the associated Energy-Based Model (EBM) via Langevin dynamics (Markowich & Villani, 2000). Similarly, in **Reinforcement Learning** Kaelbling et al. (1996) and robotics, interacting with the real world and dealing with non-differentiable physics engines often precludes the calculation of exact gradients for value functions or reward landscapes.

When the learning objective is not differentiable, a common approach is to approximate it with a smooth surrogate model $J_\theta \approx J$ and subsequently differentiate it. However, this indirect two-step process is fragile: a good scalar approximation does not guarantee a good gradient approximation, and small errors in $J_\theta$ can be amplified into invalid descent directions.

In this work, we address these challenges by directly estimating the gradient field through the minimization of

$$\mathbb{E}_p \left[ \|v_\theta(x) - \nabla J(x)\|^2 \right], \tag{1}$$

where $v_\theta(x)$ is a parameterized estimator of the gradient of $J(x)$, and $p(x)$ denotes an appropriate sampling distribution. Unlike standard surrogate models trained to approximate $J(x)$ in an $L^2$ sense, our method directly enforces fidelity in the gradient approximation. By leveraging a weak formulation of the derivative and incorporating Sobolev norms, the proposed approach is well-suited to scenarios where $J(x)$ is accessible only through scalar evaluations.

This formulation is particularly advantageous in applications where true gradients are either unavailable or computationally prohibitive to obtain. While our method is general, we illustrate its potential on a challenging problem in geosciences: the estimation of the optimal initial condition for a forecast model based on past observations. This problem, known as Data Assimilation, can be formulated as a constrained regression task:

$$\arg\min_\theta \mathbb{E}_{(x,y)\sim\mathcal{D}} \left[ \sum_{t=1}^T \|\mathcal{H}(\mathcal{M}(f_\theta(x), t)) - y_t\|_{\Sigma_t}^2 \right], \tag{2}$$

where $\mathcal{H}$ is an observational model and $\mathcal{M}$ is a numerical model (e.g., Navier-Stokes). Identifying an optimal operator $f$ without access to the adjoint of $\mathcal{M}$ represents a significant challenge where our weak gradient estimation framework proves decisive.

### 1.1 CONTRIBUTIONS

Our contributions can be stated as follows:

**Weak gradient matching:** We introduce a framework that repurposes the Generalized Score Matching identity to learn gradients of arbitrary black-box functions. To the best of our knowledge, and despite the extensive literature on Stein's method and Score Matching, the direct utilization of this identity to learn the gradient field of an arbitrary black-box function as an offline parametric regression task remains unexplored.

**Implicit learning methodology:** Building on this estimator, we introduce a methodology for learning problems where both training and inference are performed indirectly. Specifically, we employ a scheme that mimics gradient descent on the expectation of targeted loss functions. By the Girsanov theorem we identify our learning objective as the path-space equivalent of the Reverse KL divergence w.r.t Langevin stochastic processes. This effectively enables the learning of Gibbs samplers over trajectories even when the gradient of the energy is inaccessible.

**Application to scientific ML:** We demonstrate the practical utility of our approach on a high-dimensional data assimilation task involving the **two-dimensional Navier-Stokes equations**. We show that WGM can effectively solve inverse problems involving chaotic dynamics and non-differentiable numerical solvers, where traditional finite-difference baselines fail to provide valid descent directions.

## 2 Background

### 2.1 Weak Derivatives and Related Work

Let $\Omega \subset \mathbb{R}^d$ be an open domain and $J \in L^2(\Omega)$. Instead of relying on the classical pointwise derivative, we adopt the notion of weak differentiation. A function $v \in L^2(\Omega)$ is the weak gradient of $J$ if, for all smooth test functions with compact support $\varphi \in C_c^\infty(\Omega)$, the integration by parts identity holds:

$$\langle J, \nabla\varphi \rangle_{L^2} = -\langle v, \varphi \rangle_{L^2}. \tag{3}$$

This formulation is central to Scientific Machine Learning when dealing with solutions of low regularity. In the context of Physics-Informed Neural Networks (PINNs) (Raissi et al., 2017), weak formulations (wPINNs) (De Ryck et al., 2024; Chen et al., 2023) leverage this identity against a *known* PDE operator to prevent loss explosions at discontinuities. In contrast, our work applies this principle to estimate *unknown* gradient fields from black-box samples. As we show in Section 4, this allows capturing distributional derivatives (e.g., of step functions) that backpropagation cannot.

Finally, our framework opens the possibility to perform *Sobolev Training* (Czarnecki et al., 2017), *i.e.* minimizing the loss $\ell(\theta) = \|J_\theta - J\|_{L^2}^2 + \lambda\|\nabla J_\theta - \nabla J\|_{L^2}^2$, in regimes where it was previously inapplicable. While this objective normally requires ground-truth gradients $\nabla J$, our weak formulation allows us minimizing the gradient penalty term using only scalar evaluations. This effectively unlocks higher-order regularization for black-box regression problems, improving data efficiency without requiring an adjoint model.

### 2.2 Gradient Field Estimation from Scalar Evaluations

When the gradient $\nabla J$ is not accessible via backpropagation (LeCun et al., 1988) or adjoint methods (DIMET & TALAGRAND, 1986), it must be inferred from scalar evaluations $J(x)$. This task, learning a functional approximation of the gradient field, must be distinguished from the related but distinct problem of black-box optimization. Within the scope of learning a gradient field, the standard baselines are:

**Finite Differences / SPSA:** These methods provide pointwise estimates of the gradient. In high dimension, the stochastic finite difference estimator (Spall, 1997) (SPSA) requires two evaluations of $J$ per sample, where gradients are approximated as $\hat{g}(x) = \frac{J(x+\varepsilon v) - J(x-\varepsilon v)}{2\varepsilon}v$ with $v \sim \mathcal{N}(0, I_d)$. Its variance, $\sim \sigma^2/\varepsilon^2$, makes it fragile under scalar noise.

**Surrogate Modeling + Automatic Differentiation:** The most common approach for indirect gradient estimation consists in training a differentiable scalar surrogate $J_\theta$ by minimizing an $L^2$ loss

$$\min_\theta \ \mathbb{E}_p\big[\|J_\theta(x) - J(x)\|^2\big]$$

and then extracting gradients via automatic differentiation of the fitted model $\nabla J_\theta(x)$ (Conn et al., 2009). Although this two-step strategy frequently achieves excellent scalar approximation, the main weakness of this approach is that the gradients are estimated indirectly; it is the function itself that is targeted in the learning process, and not its derivatives; We prove in A.4.1 that this strategy in general non-optimal.

These indirect approaches, they remains at the core of many state-of-the-art Black-box optimization methods e.g., ZO-Adam (Chen et al., 2019),(Grathwohl et al., 2017)), or evolution strategies (Wierstra et al., 2014). These methods aim at an *iterative* search for an optimal input $x^*$ that minimizes or maximizes $J(x)$. Their output is a solution point or population, not a reusable gradient model. In contrast, our objective is *offline* learning of a functional approximation to the entire gradient field $\nabla J(x)$, which can be queried repeatedly for amortized variational inference (see 3.4).

#### 2.2.1 A particular case: gradient of log data distribution

Score matching (Hyvärinen & Dayan, 2005) is a widely used statistical learning technique for estimating the parameters of unnormalized probability densities. It minimizes the discrepancy

**Table 1:** Comparison between different approaches for black-box gradient fields estimation.

| Feature | SPSA | Surrogate-AD | WGM |
|---|---|---|---|
| Direct estimate of $\nabla J$ | ✓(pointwise) | ✗ | ✓(global) |
| Noise robustness | ✗ | ✓ | ✓ |
| Trainable from $(x_i, J(x_i))$ | ✗ | ✓ | ✓ |
| Unknown $\nabla \log p$ | ✓ | ✓ | ●(proxy) |
| Flexible parameterization | ✓ | ✗(conservative) | ✓ |

between the gradient field (or "score function") of a parametric model and that of the target distribution, thereby avoiding the often intractable computation of the normalization constant. More precisely, score matching optimizes a loss function that measures the mean squared error (MSE) between the model's estimated score and the true gradient of the log-density:

$$\mathcal{L}(\theta) = \mathbb{E}_{p_{\text{data}}(x)} \left[ \| \nabla_x \log p_\theta(x) - \nabla_x \log p_{\text{data}}(x) \|^2 \right]. \tag{4}$$

Since the true score function $\nabla_x \log p_{\text{data}}(x)$ is unknown, it is approximated using independently and identically distributed (i.i.d.) samples from the data distribution. An equivalent loss function that can be estimated directly from samples is:

$$\mathcal{L}(\theta) = \mathbb{E}_{p_{\text{data}}(x)} \left[ \frac{1}{2} \| \nabla_x \log p_\theta(x) \|^2 + \text{div}\big( \nabla_x \log p_\theta(x) \big) \right]. \tag{5}$$

This formulation allows the model to align its score function with the underlying data distribution without explicitly computing the normalization constant. More importantly, score matching can be interpreted as a weak form of gradient estimation in generative modeling. Rather than requiring direct access to the true probability density, it leverages integration by parts to estimate the gradient of the log-density in a distributional (or "weak") sense.

## 3 METHOD

### 3.1 MAIN RESULT: THE WEAK GRADIENT MATCHING COST

To address these multiple challenges and drawing inspiration from score matching (Hyvärinen & Dayan, 2005), we propose an alternative to direct mean square error minimization between a field $v_\theta$ and a targeted gradient field $\nabla J$ by introducing an equivalent learning criterion:

**Proposition 3.1.** *Let $p$ be a differentiable probability distribution defined on an open set $\Omega$ and $\|\nabla J\|$ and $\|v_\theta\|$ are square integrable under $p$. If $\lim_{x \longrightarrow \partial \Omega} p(x) J(x) = 0$,*

*then, the expectation of $\|v_\theta - \nabla J\|^2$ under $p$ verifies:*

$$E_p[\|v_\theta - \nabla J\|^2] = \mathbb{E}_p \big[ \|v_\theta\|^2 + \|\nabla J\|^2 + 2J(x)\big(div(v_\theta) + \langle v_\theta, \nabla \log p \rangle\big) \big]. \tag{6}$$

**Remark:** The proposed loss function allows us to parameterize the learnable gradient field $v_\theta$ without strictly constraining it to be the gradient of a scalar function $V_\theta$. Enforcing a conservative constraint $v_\theta = \nabla V_\theta$ appears interesting for two reasons: $\nabla V_\theta$ is already the gradient of a potential (regularizing effect) and it allow to constrain values of the potential by an additional term $\|V_\theta - J\|^2$ as well as its gradient (by 3.1). We will denote this approach by *mixed* or *Sobolev training.* In our numerical experiments, we compare the performance of several approaches: non-conservative, conservative, and variants with or without Sobolev regularization.

**proof:** See appendix A.1. **Example** Application to the quadratic case in appendix A.2.1

In the context of estimating the gradient of a scalar function, the substitution of the left-hand side by the right-hand side in the regression problem 3.1 is particularly appealing: The regression problem (w.r.t $\theta$) no longer depends on $\nabla J$, but rely instead only on scalar

evaluations $J(x)$. In what follows, we denote by $\mathcal{L}_{WGM}(\theta) = \mathbb{E}_p\big[\|v_\theta\|_2 + 2J(x)(div(v_\theta(x)) + \langle v_\theta, \nabla \log p \rangle)\big]$ the weak gradient matching cost associated to A.1. In contrast to score matching (Hyvärinen & Dayan, 2005), originally designed to recover the score function $\nabla \log(p)$, our formulation extends the integration-by-parts principle to estimate $\nabla J$ for a general scalar field J. The practical implementation of WGM relies on access to the score function $\nabla \log p$. This requirement is naturally met when working with explicit generative models providing tractable scores, such as Normalizing Flows (Rezende & Mohamed, 2015) or Energy-Based Models (LeCun et al., 2006). In the absence of such models, we show in Section 3.3 how to employ a non-parametric approximation (KDE) to provide an analytical score. **Control variate:** To further stabilize training, we exploit the invariance property $\nabla(J - c) = \nabla J$. We investigate the choice of an optimal baseline $c$ that minimizes the estimator's variance; the derivation is provided in Appendix A.2.

### 3.2 HUTCHINSON ESTIMATION OF THE DIVERGENCE

While the proposed estimator enables gradient estimation using only scalar function evaluations, this comes at the cost of computing the divergence of the estimator $v_\theta$. For a function $v_\theta : \mathbb{R}^n \to \mathbb{R}^n$, the divergence is defined as: $div(v_\theta) = \sum_{i=1}^n \frac{\partial v_\theta(x)_i}{\partial x_i}$. Computing this divergence exactly requires $n$ backpropagations per sample, making it computationally expensive for high-dimensional inputs. To alleviate this, we employ the Hutchinson Trace Estimator(Hutchinson, 1989), a stochastic approximation method that significantly reduces computational cost while remaining consistent with existing approaches. Using a single random Gaussian vector $\mathbf{h} \sim \mathcal{N}(0, I_n)$, the divergence can be approximated as follows:

$$\mathrm{div}(v_\theta) \approx \mathbf{h}^T \cdot \nabla_x\big(v_\theta(x) \cdot \mathbf{h}\big), \tag{7}$$

where the gradient $\nabla_x(v_\theta(x) \cdot \mathbf{v})$ is efficiently computed via back-propagation. This stochastic approximation has previously been employed in the matching of sliced scores (Song et al., 2020) and continuous normalization of flows (Grathwohl et al., 2018), demonstrating its effectiveness in reducing computational cost while maintaining accuracy.

#### 3.2.1 ASYMPTOTIC NORMALITY

As the divergence term is approximated using Hutchinson estimator, it is particularly interesting to study the asymptotic behavior of the proposed estimator using $M$-estimation theory. The following theorem states that when we assume consistency, the dependence of the $MC$ estimation decays linearly with the numbers of Hutchinson samples.

**Proposition 3.2.** *Let $\hat{\theta}_N := \arg\min_{\theta \in \Theta} \frac{1}{N} \sum_{i=1}^N L_{\mathrm{WGM}}(\theta, X_i)$. Under the conditions of App. A, $\hat{\theta}_N$ is consistent and:*

$$\sqrt{N}\big(\hat{\theta}_N - \theta^\star\big) \xrightarrow{d} \mathcal{N}\big(0, V\big), \qquad V = H^{-1}\big(\Sigma_{det} + \tfrac{1}{M}\Sigma_{MC}\big)H^{-1},$$

*with $H = \nabla_\theta^2 L_{\mathrm{WGM}}(\theta^\star)$. The Monte-Carlo term $\Sigma_{MC}$ arises from the Hutchinson approximation of the divergence and vanishes when the exact divergence is used $(M \to \infty)$.*

**Proof:** Appendix.

### 3.3 KERNEL DENSITY APPROXIMATION OF THE SCORE

WGM requires the score $\nabla \log p(x)$ of the sampling distribution. For cases where $p$ is unknown but samples $\{x_i\}_{i=1}^N$ are available, the score can be approximated through kernel estimator and a marginalization trick (see B.1). For a gaussian kernel (KDE) we have:

$$p(x) = \frac{1}{N} \sum_{i=1}^N \mathcal{N}(x \mid x_i, \sigma^2 I).$$

This yields exactly the same optimal $\theta^*$ as the following practical procedure (Algorithm 1):

---

**Algorithm 1** WGM with approximate GMM score

---

1: Sample data point $x_i \sim \{x_1, \ldots, x_N\}$
2: Sample $\epsilon \sim \mathcal{N}(0, I)$
3: $x = x_i + \sigma\epsilon$
4: Evaluate $J(x)$
5: Compute loss $\mathcal{L} = \|v_\theta(x)\|^2 + 2\, J(x)\Big(\mathrm{div}\, v_\theta(x) + \frac{\langle v_\theta(x), x_i - x\rangle}{\sigma^2}\Big)$     $\triangleright$ Hutchinson for div
6: Update $\theta$ via $\nabla_\theta \mathcal{L}$

---

### 3.4 Amortized variational inference with Learned Gradient Fields

Many regression problems can be solved without ever differentiating through the (or even having access to) the loss with respect to the parameters $\theta$. Consider the standard expected risk

$$\min_\theta \; \mathbb{E}_{(X,Y)}\big[\mathcal{C}(f_\theta(X), Y)\big].$$

For mean-squared error $\mathcal{C}(\hat{y}, Y) = \|\hat{y} - Y\|^2$, the optimal predictor is the conditional mean $f_{\theta^*}(x) = \mathbb{E}[Y \mid X = x]$. Crucially, this optimum can also be found by treating $\hat{y} = f_\theta(x)$ as a *control variable* and running gradient descent in output space on the *conditional* risk

$$J(\hat{y}, x) \;=\; \mathbb{E}_{Y|x}\big[\mathcal{C}(\hat{y}, Y)\big]$$

(for MSE: $J(\hat{y}, x) = \|\hat{y} - \mathbb{E}[Y \mid x]\|^2$). Because the gradient commutes with the expectation in the quadratic case, $\nabla_{\hat{y}} J(\hat{y}, x)$ can be estimated unbiasedly from i.i.d. samples of $Y \mid x$.

Our key idea is to instead learn a reusable conditional gradient field $v_\theta(\hat{y}, x) \approx \nabla_{\hat{y}} J(\hat{y}, x)$ using Weak Gradient Matching on pairs $(\hat{y}, x)$ with scalar labels $J(\hat{y}, x)$. After training, inference is simply gradient descent in output space (Algorithm 2) — fully bypassing any need for backpropagation through the original model or loss.

---

**Algorithm 2** Amortized inference at test time

---

**Require:** input $x$, initial guess $\hat{y}^{(0)}$, trained field $v_\theta$, step size $\eta$, iterations $T$
1: **for** $t = 1$ **to** $T$ **do**
2:     $\hat{y}^{(t)} \leftarrow \hat{y}^{(t-1)} - \eta\, v_\theta(\hat{y}^{(t-1)}, x)$
3: **end for**
4: **return** $\hat{y}^{(T)}$

---

This scheme naturally extends to non-quadratic losses where unbiased gradient estimation is still possible (e.g., cross-entropy with sampled labels). In the next section we introduce the learning methodology used for efficient amortized variational inference and establish a link between these methodology and reverse KL on path spaces.

### 3.5 Mimicking gradient descent learning scheme

To effectively optimize the targeted cost function, we propose learning the gradient estimator jointly with the gradient descent trajectories. This approach aims to minimize the discrepancy between the learned field $v_\theta$ and the true gradient $\nabla J$ specifically along the optimization path. Formally, we define the sampling distribution $p_{\theta_k}(x)$ as the marginal of trajectories generated by the previous model iteration $\theta_k$, taking the form $p_{\theta_k}(x) = \int p(x|t)p(t)dt$, where $p(x|t)$ is centered on the current state estimate $V_{\theta_k}(t)$. The learning objective at step $k$ consists of minimizing the

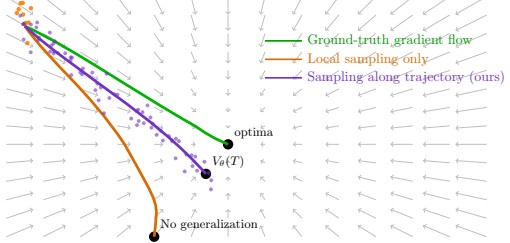

**Figure 1: Learning scheme:** Unlike static strategies (orange), the mimicking gradient scheme (purple) learns the true gradient path (green) by sampling along the trajectory. This dynamic sampling ensures better generalization for amortized inference.

expected squared error over these trajectories:

$$\mathcal{L}(\theta, \theta_k) = \mathbb{E}_{x \sim p_{\theta_k}} \left[ \|v_\theta(x) - \nabla J(x)\|^2 \right]. \tag{8}$$

Since the true gradient $\nabla J$ is unknown, we invoke Proposition A.1 (WGM) to rewrite this intractable objective into a computable loss relying solely on scalar evaluations of $J$:

$$\mathcal{L}(\theta, \theta_k) \cong \frac{1}{T} \int_0^T \mathbb{E}_{x \sim p_{\theta_k}(\cdot|t)} \left[ \|v_\theta(x)\|_2^2 + 2J(x)\big(\mathrm{div}(v_\theta(x)) + \langle v_\theta(x), \nabla_x \log p(x|t) \rangle \big) \right]. \tag{9}$$

The trajectory distribution is obtained via the numerical integration of the ODE associated with the Cauchy problem:

$$\begin{cases} V'_{\theta_k}(t) = -v_{\theta_k}\big(V_{\theta_k}(t)\big), & t \in [0, T] \\ V_{\theta_k}(0) = V_0. \end{cases} \tag{10}$$

Once the trajectories are integrated, we minimize the weak gradient matching cost by sampling around the states $V_{\theta_k}(s)$. The learning procedure follows the iteration $\theta_{k+1} = \arg_\theta \min \mathcal{L}(\theta, \theta_k)$, as summarized in Algorithm 3.

### 3.5.1 Theoretical Interpretation: Path-Space Reverse KL

In settings where no data samples are available, learning a sampler typically requires minimizing the Reverse KL divergence. However, standard approaches minimize this divergence on the *marginal density* at final $T$. This strategy is well-known to suffer from mode collapse, failing to cover multimodal targets (Gabrié et al., 2022). Our framework addresses this by lifting the objective to the **path space**. According to Girsanov's theorem, the KL divergence between the trajectories generated by our model ($\mathbb{P}_\theta$) and the true Langevin dynamics ($\mathbb{Q}$) is proportional to the integrated gradient error:

$$\mathrm{KL}(\mathbb{P}_\theta \| \mathbb{Q}) \propto \mathbb{E}_{\mathbb{P}_\theta} \left[ \int_0^T \|v_\theta(X_t) - \nabla J(X_t)\|^2 dt \right]. \tag{11}$$

Our Weak Gradient Matching (WGM) formulation allows us to minimize this integral without ever accessing the true gradient $\nabla J$. By doing so, we perform a particular instance of **Gradient Flow Matching** (Lipman et al., 2022): we constrain the estimator to match the targeted gradient dynamics along the entire optimization path.

## 4 Numerical experiments

### 4.1 Numerical experiments under exact score distribution

#### 4.1.1 Representation of weak derivatives.

The concept of a weak derivative extends the classical notion of differentiation to functions that are not differentiable in the traditional sense. A canonical example is the derivative of the indicator function $\mathbb{1}_{[a,b]}$, which in the distributional sense satisfies $\partial_x \mathbb{1}_{[a,b]} = \delta_a - \delta_b$. Directly matching such weak derivatives in functional spaces commonly used in machine learning, such as neural networks or RKHS (Paulsen & Raghupathi, 2016), is generally infeasible. For instance, in an RKHS with kernel $K$, we obtain the approximation $\delta_a - \delta_b \approx K(\cdot, a) - K(\cdot, b)$, which is not an exact representation. Consequently, fitting a model to the weak derivative $\partial_x \mathbb{1}_{[a,b]}$ using direct gradient matching is impossible in a strict sense, since standard backpropagation yields $\partial_x \mathbb{1}_{[a,b]} = 0$ almost everywhere due to the function's discontinuity. However, as illustrated in Figure 2, the proposed weak gradient cost enables us to learn a neural network that approximates the distributional derivative of $\mathbb{1}_{[a,b]}$, capturing the jump discontinuities at the boundaries $a$ and $b$ through the structure of the cost, rather than through point-wise gradients.

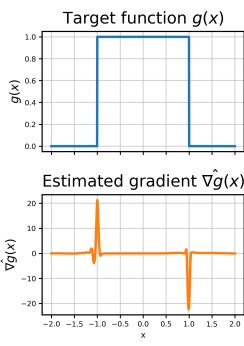

**Figure 2:** Left: function $f = \mathbb{1}_{[-1,1]}$, right: neural network approximation of $\frac{\partial f}{\partial x}$ in the weak sense.

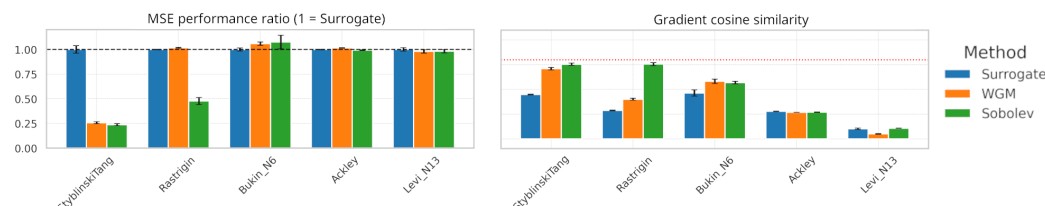

**Figure 3:** Top: comparison of normalized MSE (Surrogate = 1) for different methods (lower is better). Bottom: cosine similarity between learned and target gradient fields (Higher is better).

### 4.1.2 Linear regression on Gaussian kernels

As the weak gradient matching cost is convex and involves linear operators, it is particularly well-suited for estimators that are linear in their parameters, $v_\theta(x) = \sum_i \theta_i \phi_i(x)$. This class of models includes finite element methods (Reddy, 1993), polynomial regression (Ostertagová, 2012), and Radial Basis Function (RBF) networks, such as those using Gaussian kernels. For such parameterizations, the optimization problem can be solved optimally and the solution given explicitly. Figure 3 compares different strategies using an identical Gaussian kernel setup. The results show that minimizing different objectives can lead to distinct gradient approximations. While surrogate models aim for functional fidelity, their derivatives are not always accurate. In contrast, WGM directly optimizes an objective tied to the gradient error, which can yield a more faithful gradient field estimation, as reflected by the mean squared error and cosine similarity metrics.

### 4.1.3 Experiment with neural networks

**Table 2:** Normalized mean square error for different targeted gradient fields & methods.

| Target Family | Problem Setup | | Normalized MSE ± std | | | |
| | Dim. | Noise | WGM-C | WGM-NC | Sobolev | SPSA |
|---|---|---|---|---|---|---|
| Gibbs Energies | $16 \times 16$ | 0.0 | $0.89 \pm 0.29$ | $0.92 \pm 0.30$ | $0.87 \pm 0.11$ | $\mathbf{0.13 \pm 0.05}$ |
| | | 0.1 | $\mathbf{0.78 \pm 0.21}$ | $0.80 \pm 0.32$ | $0.80 \pm 0.09$ | $6.76 \pm 2.09$ |
| Random CNNs | $16 \times 16$ | 0.0 | $\mathbf{0.78 \pm 0.47}$ | $1.45 \pm 0.97$ | $0.80 \pm 0.09$ | $0.89 \pm 0.47$ |
| | | 0.1 | $\mathbf{0.76 \pm 0.24}$ | $1.54 \pm 0.55$ | $0.86 \pm 0.09$ | $1.65 \pm 1.09$ |

For benchmarking purposes[1], we evaluate how each estimator scales with input dimension and responds to variance from Hutchinson trace estimation. The targeted gradient fields are:

**Random targets:** $\nabla J : \mathbb{R}^d \to \mathbb{R}^d$ are generated by random initializations of a CNN. Since methods targeting log-density scores—such as classical score matching or Stein-kernel estimators—require assumptions incompatible with general scalar black-box settings, they are excluded from comparison.

**Gibbs energies:** We estimate the score of different Gibbs energies, such as Ising models, from samples arising from another distribution (a task impossible for implicit score matching (Hyvärinen & Dayan, 2005; Song et al., 2020)).

**Considered baselines:** The benchmarks are listed below.

**WGM**, our proposed weak-gradient matching method using 5 Hutchinson vectors per sample; **Sobolev**, a hybrid approach (Czarnecki et al., 2017) that combines surrogate modeling with an added **WGM** regularization term; **SPSA**, the Simultaneous Perturbation Stochastic Approximation; and **Surrogate-AD**, which trains a neural network proxy for $J(\cdot)$ using an $L_2$ loss and differentiates it via autodiff. For each considered method, we perform a grid search over hyperparameters (learning rates, perturbation size $\epsilon$).

---

[1]Experiments can be reproduced using the anonymized repository: `https://anonymous.4open.science/r/WGM-850D`

## 4.2 Learning from a score approximation

We investigate the scenario where the true score is unknown and replaced by an approximation (learning with the "wrong" score). We validate this approach on MNIST using two distinct target functions $J$: (i) MSE for $4\times$ super-resolution, and (ii) cross-entropy for even/odd digit classification. A super resolution experiment is also performed on CIFAR-10. As shown in Table 3, even with a naïve approximation of data density via KDE, the proposed approach outperforms surrogate models for gradient estimation. On the super-resolution task, we illustrate the sensitivity of the method to the kernel bandwidth $\sigma$ in Figure 4. Additionally, we provide an ablation study on the number of Hutchinson vectors in Appendix C.1.1, demonstrating the method's robustness to stochastic divergence estimation which is in line with (Song et al., 2020; Grathwohl et al., 2018).

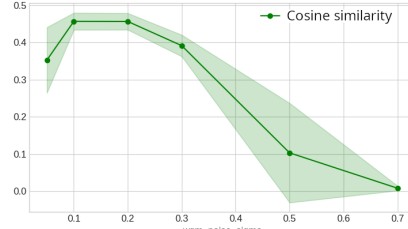

**Figure 4:** Cosine similarity between oracle gradient and WGM. The method is sensitive to $\sigma$: we advise conducting MLE to tune this parameter.

**Table 3:** Evaluation metrics w.r.t oracle gradients. Interestingly our approaches remains competitive with SPSA while surrogate models fails to capture correct gradient direction once differentiated as shown by the cosine metrics.

| Dataset | Target ($J$) | Metric | WGM | WGM-NC | Surrogate | SPSA |
|---|---|---|---|---|---|---|
| MNIST | Super Res. | **MSE** | $0.52 \pm 7\mathrm{e}{-3}$ | $0.47 \pm 2\mathrm{e}{-4}$ | $0.64 \pm 2\mathrm{e}{-3}$ | $\mathbf{0.43 \pm 3e{-4}}$ |
| | | **Cos** | $0.43 \pm 9\mathrm{e}{-3}$ | $0.51 \pm 6\mathrm{e}{-3}$ | $0.006 \pm 8\mathrm{e}{-3}$ | $\mathbf{0.57 \pm 2e{-4}}$ |
| | Classifier | **MSE** | $\mathbf{5.8e{-3} \pm 7e{-5}}$ | $6.2\mathrm{e}{-3} \pm 1.6\mathrm{e}{-5}$ | $6.6\mathrm{e}{-3} \pm 3\mathrm{e}{-7}$ | $5.9\mathrm{e}{-3} \pm 3\mathrm{e}{-7}$ |
| | | **Cos** | $\mathbf{0.315 \pm 0.035}$ | $0.23 \pm 5\mathrm{e}{-2}$ | $0.14 \pm 3\mathrm{e}{-3}$ | $0.305 \pm 8\mathrm{e}{-3}$ |
| CIFAR10 | Super Res. | **MSE** | $1.56\mathrm{e}{-2} \pm 2.4\mathrm{e}{-4}$ | $\mathbf{1.47e{-2} \pm 2.3e{-4}}$ | $3.0e-2 \pm 7.5e-3$ | $1.55\mathrm{e}{-2} \pm 5.5\mathrm{e}{-4}$ |
| | | **Cos** | $0.54 \pm 1.2\mathrm{e}{-2}$ | $0.61 \pm 2.3\mathrm{e}{-2}$ | $8.4e-3 \pm 1.4e-2$ | $\mathbf{0.63 \pm 4e{-2}}$ |

## 4.3 Addressing data assimilation task with non differentiable code

We now apply the proposed methodology to predict gradient descent paths associated to a non differentiable variational cost involving the numerical integration of Navier-Stokes equations. The next section introduce the context of data assimilation.

### 4.3.1 Data Assimilation & The 4D-VAR Algorithm

Data assimilation (Evensen, 2009) is a research field that focuses on state estimation problem in geosciences using observational data from remote sensing or in-situ probes. The goal is to infer the true state of a physical system from partial and noisy observations. Formally, given an observational model $\mathcal{H}$ that relates the true system state $X$ to the observations $Y$ via

$$Y = \mathcal{H}(X) + \varepsilon,$$

where $\varepsilon$ represents observational noise $\sim \mathcal{N}(0, \Sigma_t)$, the problem is often formulated as an ill-posed inverse problem that requires regularization to obtain a meaningful solution. A prominent method in data assimilation is the Four-Dimensional Variational (4D-VAR) algorithm. In its strong constraint formulation (Evensen, 2009; Talagrand & Courtier, 1987) referred to as *SC-4DVAR*, the numerical model is assumed to be perfect. The algorithm aims to find the initial condition that minimizes the forecast error, given by the Mahanalobis $L^2(\Omega)$ norm, with respect to $\Sigma_i$:

$$J_{SC}(x_0) = \sum_{i=0}^{T} \|\mathcal{H}_i(\mathcal{M}(x_0, t_i) - y_i)\|_{\Sigma_i}^2.$$

Although 4D-VAR offers a principled framework for variational inference, it is often impractical in real-world settings. Many physical models, such as NEMO (Madec et al., 2015), rely

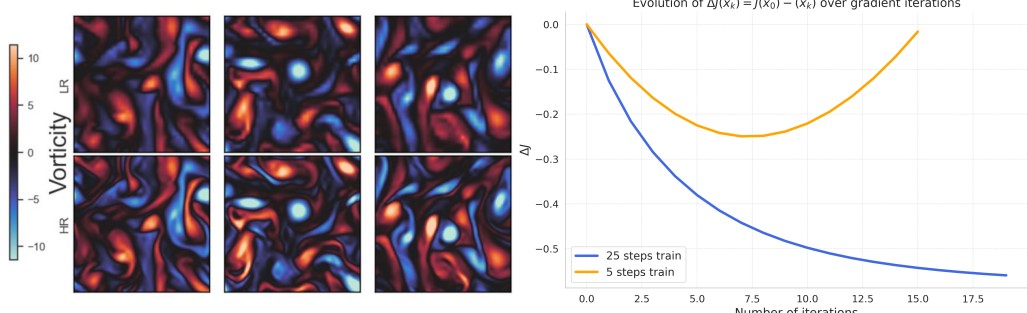

**Figure 5: (Left)** Initial conditioning input and the estimated states after performing numerical integration over the predicted gradient field. **(Right)** Comparison for networks trained to mimic 5 (resp. 25) gradient steps. As more steps are performed during training, the expected variational cost is minimized more effectively, which validates the proposed methodology.

on legacy Fortran code and do not include an adjoint implementation. In other cases, the evolution operator is non-differentiable (e.g., sea ice dynamics), or governed by stochastic differential equations (SDEs) for which no scalable high-dimensional gradient estimation technique exists. In the next section we illustrate the application of the weak gradient matching estimator in such non differentiable context.

### 4.3.2 LEARNING FROM A NON-DIFFERENTIABLE PHYSICS SOLVER: APPLICATION TO 2D NAVIER–STOKES

We generate ensembles of trajectories of the incompressible 2D Navier–Stokes equations using the pseudo-spectral solver from `torch-cfd`[2] on a $128 \times 128$ periodic grid. Random divergence-free velocity fields are used as initial conditions. We cast the task as a strong-

**Table 4:** Change in 4DVAR cost (lower is better).

| Method | WGM | | Surr. | SPSA |
|--------|-----|-----|-------|------|
| Res. | $16^2$ | $32^2$ | $16^2$ | $16^2$ |
| $\Delta$ Cost | **-0.72** | -0.51 | +0.02 | -0.08 |

constraint 4DVAR assimilation problem: given partial observations in the time window $[0.1, 0.5]$, recover the initial condition $u(0)$ by minimizing the 4DVAR cost, defined as the $\ell^2$ reconstruction error restricted to the observed subdomain. We adopt a patch-based scheme: the cost is computed locally on non-overlapping spatial patches of size $K \times K$ (with $K \in \{16, 32\}$), reducing dimensionality and computational burden. We train a three-layer CNN gradient estimator $v_\theta$ with different learning objective and use it within a learned gradient descent procedure. As illustrated in Figure 5, networks trained to follow longer optimization horizons achieve more stable and effective minimization of the 4DVAR cost. For baselines, we reuse the same CNN architecture within the amortized-inference framework but replace the WGM objective with either (i) a surrogate cost regression differentiated via autodiff, or (ii) a finite-difference (SPSA) estimator. As shown in Table 4, only WGM reduces the true 4DVAR objective: we hypothesise that low perturbation of the initial state are dissipated by the numerical model which make the finite difference approach fails. This confirms that explicit gradient learning is necessary for valid inference.

### 4.4 CONCLUSION

In this work, we introduce a new data-driven method for estimating function gradients in non-differentiable settings. This connects the problem of derivative estimation with techniques known in score matching opening an interesting new research direction. Our results demonstrate the effectiveness of this approach even in complex scenarios, such as Navier-Stokes dynamics and nonlinear data assimilation. The method shows promise as a robust and cost-effective alternative to both random finite differences and surrogate models, offering improved noise tolerance and reduced reliance on expensive function evaluations.

---

[2]`https://github.com/scaomath/torch-cfd`

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

APPENDIX / SUPPLEMENTAL MATERIAL

## CONTENTS

## A    Proofs of the main results

### A.1    Derivation of the weak gradient matching cost

**Proposition A.1.** *Let $p$ be a differentiable probability distribution such that $pJ = 0$ on $\partial\Omega$ and let $J$ be a differentiable scalar potential, then, the expectation of $\|v_\theta - \nabla J\|^2$ under $p$ verifies:*

$$E_p[\|v - \nabla J\|^2] = \mathbb{E}_p\big[\|v\|^2 + \|\nabla J\|^2 + 2J(x)\big(div(v) + \langle v, \nabla \log p\rangle\big)\big]. \tag{12}$$

**proof:**

We develop the square norm:

$$\mathbb{E}_p\|v - \nabla J\|^2 = \mathbb{E}_p\big[\|v\|^2 + \|\nabla J\|^2 - 2\langle v(x), \nabla J\rangle\big].$$

As the targeted optimization issue does not depends on $\nabla J$, we can focus on the third term:

$$\mathbb{E}_p\big[\langle v, \nabla J\rangle\big] = \int_\Omega \langle p(x)v(x), \nabla J(x)\rangle dx.$$

The divergence formula leads to:

$$= \int_{\partial\omega} J(x)p(x)v(x) \cdot n ds - \int_\Omega J(x)div\big(v(x)p(x)\big)dx$$

If the sampling distribution is chosen such as $p_{|\partial\Omega} = 0$ or $J_{|\partial\Omega} = 0$, the second right-hand side term above equals to:

$$\int_\Omega J(x)\big[div\big(v(x)\big)p(x) + \langle v(x), \nabla p(x)\rangle\big]dx.$$

Choosing $p(x) = K\exp\big(f(x)\big)$, then $\nabla p(x) = p(x)\nabla f(x)$ and we have:

$$\int_\Omega J(x)\big[div(v(x))p(x) + \langle v(x), \nabla p(x)\rangle\big]dx = \mathbb{E}_p\big[J(x)\big(div(v(x)) + \langle v(x), \nabla f(x)\rangle\big)\big],$$

which completes the proof.

As the above proposition may seem restrictive, the following lemma extends the results to unbounded domain when the product $p(x)J(x)$ vanishes when $|x|$ grows.

**Lemma A.2.** *Let $\Omega \subset \mathbb{R}^n$ be an unbounded domain. Proposition A.1 remains valid provided $J$ is square integrable under $p$ : $\mathbb{E}_p[\|J(x)\|^2] < +\infty$*

**Proof:**

Let $B(0, 2R)$, the ball of radius $2R$ with $R$ strictly positive and a truncated function $\Phi_R(x) \in \mathcal{C}^1(\Omega, \mathbb{R})$:

- $\Phi_R(x) = 1$ on $B(0, R)$.
- There exist $C$ a positive constant such that $|\nabla\Phi_R(x)| \leq \frac{C}{R}$ for $R \leq \|x\| \leq 2R$.
- $\Phi_R(x) = 0$ for $\|x\| \geq 2R$.

Then, the condition A.1 holds with $J_R = J\Phi_R$. We have:

$$\mathbb{E}_p\big[\langle v, \nabla J_R\rangle\big] = \mathbb{E}_p\big[J_R(x)\big(div(v(x)) + \langle v(x), \nabla f(x)\rangle\big)\big] \tag{13}$$

For any $x$ we have the point-wise convergence $J(x) = \lim_{\|R\|\to\infty} J_R(x)$. We will now show that the the expression 13 is bounded by an integrable function to apply the dominated convergence theorem:

$$\mathbb{E}_p\big[|\langle v, \nabla J_R\rangle|\big] \leq \mathbb{E}_p\big[|\langle v, \Phi_R \nabla J\rangle|\big] + \mathbb{E}_p\big[|J(x)\langle v, \nabla \Phi_R\rangle|\big]$$

As $\Phi_R \leq 1$ The first term is bounded by $E_p\big[|\langle v, \nabla J\rangle|\big]$. For the second term we have:

$$\mathbb{E}_p\big[|J(x)\langle v, \nabla \Phi_R\rangle|\big] \leq \frac{C}{R}\sqrt{\mathbb{E}_p\big[\|J(x)\|^2\big]\mathbb{E}_p\big[\|v(x)\|^2\big]} < +\infty$$

By the dominated convergence theorem we have:

$$\lim_{R\to\infty} \mathbb{E}_p\big[\langle v, \nabla J_R\rangle\big] = \mathbb{E}_p\big[\lim_{R\to\infty}\langle v, \nabla J_R\rangle\big] = \mathbb{E}_p\big[J(x)\big(div(v(x)) + \langle v(x), \nabla f(x)\rangle\big)\big]$$

$\square$

## A.2 Variance Reduction using Control Variates

The gradient of the loss is invariant to a constant shift in the function, i.e., $\nabla J(\mathbf{x}) = \nabla(J(\mathbf{x}) - c)$. It makes sense to center the function $J(\mathbf{x})$, but a more efficient approach to variance reduction is to use a control variate.

Let $H_\theta(\mathbf{x}) = \text{div}(v_\theta(\mathbf{x})) + \langle v_\theta(\mathbf{x}), \nabla \log p(\mathbf{x})\rangle$. A key property of this term is that its expectation is zero, $\mathbb{E}_p[H_\theta(\mathbf{x})] = 0$, under the same conditions as Proposition 3.1. We can therefore use it as a control variate for the stochastic term $J(\mathbf{x})H_\theta(\mathbf{x})$ in the loss function, replacing it with $J(\mathbf{x})H_\theta(\mathbf{x}) - cH_\theta(\mathbf{x})$ without changing the objective's expectation.

*Proof.* Let the variance of the control variate estimator be denoted by $V(c)$:

$$V(c) = \text{Var}_p\big[J(\mathbf{x})H_\theta(\mathbf{x}) - cH_\theta(\mathbf{x})\big]$$

Our goal is to find the value of $c$ that minimizes $V(c)$. We can expand the variance expression as follows:

$$V(c) = \mathbb{E}_p\left[\Big(\big(J(\mathbf{x})H_\theta(\mathbf{x}) - cH_\theta(\mathbf{x})\big) - \mathbb{E}_p\big[J(\mathbf{x})H_\theta(\mathbf{x}) - cH_\theta(\mathbf{x})\big]\Big)^2\right]$$

$$= \mathbb{E}_p\left[\Big(\big(J(\mathbf{x})H_\theta(\mathbf{x}) - \mathbb{E}_p[J(\mathbf{x})H_\theta(\mathbf{x})]\big) - c\big(H_\theta(\mathbf{x}) - \mathbb{E}_p[H_\theta(\mathbf{x})]\big)\Big)^2\right]$$

$$= \mathbb{E}_p\left[\big(J(\mathbf{x})H_\theta(\mathbf{x}) - \mathbb{E}_p[J(\mathbf{x})H_\theta(\mathbf{x})]\big)^2\right]$$

$$\quad - 2c\,\mathbb{E}_p\left[\big(J(\mathbf{x})H_\theta(\mathbf{x}) - \mathbb{E}_p[J(\mathbf{x})H_\theta(\mathbf{x})]\big)\big(H_\theta(\mathbf{x}) - \mathbb{E}_p[H_\theta(\mathbf{x})]\big)\right]$$

$$\quad + c^2\,\mathbb{E}_p\left[\big(H_\theta(\mathbf{x}) - \mathbb{E}_p[H_\theta(\mathbf{x})]\big)^2\right]$$

$$= \text{Var}_p\big(J(\mathbf{x})H_\theta(\mathbf{x})\big) - 2c\,\text{Cov}_p\big(J(\mathbf{x})H_\theta(\mathbf{x}), H_\theta(\mathbf{x})\big) + c^2\,\text{Var}_p\big(H_\theta(\mathbf{x})\big)$$

This expression for $V(c)$ is a quadratic function of $c$. To find the minimum, we take the derivative with respect to $c$ and set it to zero:

$$\frac{dV(c)}{dc} = -2\,\text{Cov}_p\big(J(\mathbf{x})H_\theta(\mathbf{x}), H_\theta(\mathbf{x})\big) + 2c\,\text{Var}_p\big(H_\theta(\mathbf{x})\big)$$

Setting the derivative to zero yields the optimal constant $c^*$:

$$-2\,\text{Cov}_p\big(J(\mathbf{x})H_\theta(\mathbf{x}), H_\theta(\mathbf{x})\big) + 2c^*\,\text{Var}_p\big(H_\theta(\mathbf{x})\big) = 0$$

$$2c^*\,\text{Var}_p\big(H_\theta(\mathbf{x})\big) = 2\,\text{Cov}_p\big(J(\mathbf{x})H_\theta(\mathbf{x}), H_\theta(\mathbf{x})\big)$$

$$c^* = \frac{\text{Cov}_p\big(J(\mathbf{x})H_\theta(\mathbf{x}), H_\theta(\mathbf{x})\big)}{\text{Var}_p\big(H_\theta(\mathbf{x})\big)}$$

This concludes the proof. $\square$

### A.2.1 An example of application in the quadratic case

If $J(\mathbf{x}) = \frac{1}{2}\mathbf{x}^\top H\mathbf{x}$, the loss function $\mathcal{L}$ associated with equation A.1 with $v(\mathbf{x}) = A\mathbf{x}$ and $p = \mathcal{N}(0, I)$ is:

$$\mathcal{L}(A) = \mathbb{E}_p\big[\|A\mathbf{x}\|^2\big] + 2\,\mathbb{E}_p\big[(\mathbf{x}^\top H\mathbf{x})\big(\operatorname{tr}(A) - (A\mathbf{x})^\top\mathbf{x}\big)\big] + \text{cst}.$$

Using $\mathbb{E}_p[\mathbf{x}^\top H\mathbf{x}] = \operatorname{tr}(H)$ and $\mathbb{E}_p[(\mathbf{x}^\top H\mathbf{x})(\mathbf{x}^\top A\mathbf{x})] = \operatorname{tr}(H)\operatorname{tr}(A) + 2\operatorname{tr}(HA)$, we obtain:

$$\mathcal{L}(A) = \text{const} + \operatorname{tr}(A^\top A) + 2\operatorname{tr}(A)\operatorname{tr}(H) - 2\Big(\operatorname{tr}(H)\operatorname{tr}(A) + 2\operatorname{tr}(HA)\Big)$$

$$= \text{const} + \operatorname{tr}(A^\top A) - 4\operatorname{tr}(HA).$$

Finally, completing the square yields:

$$\mathcal{L}(A) = \text{const} + \|A - 2H\|_F^2.$$

### A.3 Asymptotic normality

We conclude this section with the asymptotic normality of the WGM estimator. In contrast to standard variational inference, our asymptotic variance accounts for Monte Carlo noise arising from the Hutchinson trace estimator used to approximate divergence terms.

**Proposition A.3** (Asymptotic Normality). *Let $\hat{\theta}_N := \arg\min_{\theta\in\Theta} \frac{1}{N}\sum_{i=1}^N L_{\mathrm{WGM}}(\theta, X_i)$ be the WGM estimator. Under the conditions of Appendix A, the estimator $\hat{\theta}_N$ is consistent and satisfies:*

$$\sqrt{N}\big(\hat{\theta}_N - \theta^\star\big) \xrightarrow{d} \mathcal{N}\big(0, V\big), \qquad V = H^{-1}\big(\Sigma_{det} + \tfrac{1}{M}\Sigma_{MC}\big)H^{-1},$$

*where $H = \nabla_\theta^2 L_{\mathrm{WGM}}(\theta^\star)$. The term $\Sigma_{MC}$ arises from the Monte-Carlo (Hutchinson) approximation of the divergence and vanishes in the limit $(M\to\infty)$, i.e., when the exact divergence is used.*

#### Consistency

We assume consistency of $\hat{\theta}_N$, which under general conditions follows from standard M-estimation theory. To prove consistency rigorously, one typically verifies the following:

- The risk function $\mathcal{L}(\theta)$ admits a unique minimizer $\theta^*$, providing a well-defined target for the estimation.
- The parameter space $\Theta$ is compact, ensuring the existence of a minimizer and controlling the behavior of the empirical loss $\mathcal{L}_N(\theta)$.
- The empirical loss $\mathcal{L}_N(\theta)$ converges uniformly to $\mathcal{L}(\theta)$ as $N\to\infty$, ensuring that $\hat{\theta}_N \to \theta^*$.

#### Asymptotic Normality

To derive the asymptotic normality of $\theta^*$ we let $\theta^*$ denote the minimizer of the risk function $\mathcal{L} = \mathbb{E}L_{\mathrm{WGM}}$. Since $\theta_N$ minimizes $\mathcal{L}_N$ we have:

$$\nabla_\theta\mathcal{L}_N(\theta_N) = 0.$$

A Taylor expansion of $\nabla_\theta\mathcal{L}_N$ around $\theta^*$ yields:

$$\nabla_\theta\mathcal{L}_N(\theta^*) = \nabla_\theta^2\mathcal{L}_N(\theta^*)(\theta^* - \theta_N) + O(\|\theta_N - \theta^*\|^2).$$

Letting $G_N = \sqrt{N}\nabla_\theta\mathcal{L}_N(\theta^*)$ and $H_N = \nabla_\theta^2\mathcal{L}_N(\theta^*)$, we observe:

- By the Central Limit Theorem, $G_N \xrightarrow{d} \mathcal{N}(0, \Sigma)$, where $\Sigma = \operatorname{Var}(\nabla_\theta\mathcal{L}(X, \theta^*))$.
- By the Law of Large Numbers, $H_N \xrightarrow{p} H = \mathbb{E}[\nabla_\theta^2\mathcal{L}(X, \theta^*)]$.

By Slutsky's theorem, we obtain:

$$\varepsilon_N = (\theta^* - \theta_N) \approx H_N^{-1}G_N \xrightarrow{d} H^{-1}G.$$

which shows that $\varepsilon_N$ converges in law to $\mathcal{N}(0, H^{-1}\Sigma H^{-1})$.

THE COVARIANCE STRUCTURE $\Sigma$

The empirical loss functional takes the form:

$$\mathcal{L}_N(\theta, x, h^M) = \frac{1}{N} \sum_{i=1}^{n} \left( \left\| v_\theta(x) \right\|^2 + 2J(x) \left[ \left\langle v_\theta(x), \nabla_x \log p(x) \right\rangle + \frac{1}{M} \sum_{k=1}^{M} \left\langle h_k, \nabla_x\big(v_\theta(x)\, h_k\big) \right\rangle \right] \right) \tag{14}$$

To simplify notation, we define the following terms:

$$I_1 = \left\| v_\theta(x) \right\|^2 + J(x) \left\langle v_\theta(x), \nabla_x \log p(x) \right\rangle, \tag{15}$$

$$I_2 = J(x)\, \mathrm{div}\, v_\theta(x), \tag{16}$$

$$\hat{I}_2 = \frac{J(x)}{M} \sum_{k=1}^{M} \left\langle h_k, \nabla_x\big(h_k\, v_\theta(x)\big) \right\rangle. \tag{17}$$

The term $I_1$ gathers the deterministic terms, $I_2$ the divergence, and $\hat{I}_2$ is the Monte-Carlo estimate of $I_2$ over the random directions $\{h_k\}$.

Then at $\theta^*$ the gradiant loss variance is

$$\Sigma = \mathrm{Var}\big(\nabla_\theta \mathcal{L}_N(\theta^*)\big)$$

$$= \mathbb{E}_p\Big[ \mathbb{E}_{p_v}\big[ \nabla_\theta(I_1 + \hat{I}_2)\, \nabla_\theta(I_1 + \hat{I}_2)^\top \big] - \Big( \mathbb{E}_{p_v} \nabla_\theta(I_1 + \hat{I}_2) \Big)\Big( \mathbb{E}_{p_v} \nabla_\theta(I_1 + \hat{I}_2) \Big)^\top \Big]. \tag{18}$$

As at $\theta^*$ the second term equals to 0, expanding the above expression leads to:

$$\Sigma = \mathbb{E}_p \mathbb{E}_{p_v} \Big[ (\nabla_\theta I_1)(\nabla_\theta I_1)^\top + (\nabla_\theta I_1)(\nabla_\theta \hat{I}_2)^\top + (\nabla_\theta \hat{I}_2)(\nabla_\theta I_1)^\top + (\nabla_\theta \hat{I}_2)(\nabla_\theta \hat{I}_2)^\top \Big] \tag{19}$$

$$= \mathbb{E}_p \Big[ (\nabla_\theta I_1)(\nabla_\theta I_1)^\top + (\nabla_\theta I_1)\, \mathbb{E}_{p_h}\big[ \nabla_\theta \hat{I}_2 \big]^\top + \mathbb{E}_{p_h}\big[ \nabla_\theta \hat{I}_2 \big]\, (\nabla_\theta I_1)^\top + \mathbb{E}_{p_h}\big[ (\nabla_\theta \hat{I}_2)(\nabla_\theta \hat{I}_2)^\top \big] \Big] \tag{20}$$

$$= \mathbb{E}_p \Big[ (\nabla_\theta I_1)(\nabla_\theta I_1)^\top + \nabla_\theta I_1\, (\nabla_\theta I_2)^\top + \nabla_\theta I_2\, (\nabla_\theta I_1)^\top + \mathbb{E}_{p_h}\big[ (\nabla_\theta \hat{I}_2)(\nabla_\theta \hat{I}_2)^\top \big] \Big]. \tag{21}$$

Hence three contributions emerge: (i) the variance of $I_1$; (ii) a cross-covariance with $I_2$; (iii) the variance of the Monte-Carlo term $\hat{I}_2$.

EXPLICIT FORM OF OF THE MONTE CARLO CONTRIBUTION

$$\mathbb{E}_{p_h}\big[ (\nabla_\theta \hat{I}_2)(\nabla_\theta \hat{I}_2)^\top \big] = \frac{J(x)^2}{M^2} \sum_{i,j=1}^{M} \mathbb{E}_{p_h} \Big[ \nabla_\theta \big\langle h^i, \nabla_x(h^i v_\theta(x)) \big\rangle\, \nabla_\theta \big\langle h^j, \nabla_x(h^j v_\theta(x)) \big\rangle^\top \Big]. \tag{22}$$

As $\nabla_x(h^{j^T} v_\theta) = D h^j$ with $D \in \mathbb{R}^{d \times d}$ the Jacobian matrix of $v_\theta$, we have:

$$(D h^j)_p = \sum_{l=1}^{d} D_{pl} h_l^i,$$

and :

$$\langle h^i, D h^i \rangle = \sum_{p=1}^{d} h_p^i (D h^i)_p.$$

We obtain thus:

$$\langle h^i, D h^i \rangle \langle h^j, D h^j \rangle = \sum_{p,k} \sum_{l,m} h_p^i h_l^i h_k^j h_m^j D_{km} D_{pl}.$$

Taking the expectation under $p(h^1, ..., h^M) = \prod_{i=1}^{M} p(h^i)$, the distribution of the sampled vectors $h^i$ with $h^i$ i.i.d :

$$\mathbb{E}_p\Big[\sum_{p,k}\sum_{l,m}h_p^i h_l^i h_k^j h_m^j D_{km}D_{pl}\Big] = \sum_{p,k}\sum_{l,m}\mathbb{E}_p\big[h_p^i h_l^i h_k^j h_m^j\big]D_{km}D_{pl}$$

First, second and fourth-order moment of $\mathcal{N}(0,1)$ gives:

$$\mathbb{E}_p\big[h_p^i h_l^i h_k^j h_m^j\big] = \mathbb{E}\big[h_p^i h_l^i h_k^j h_m^j\big] = \begin{cases} \delta_{pl}\,\delta_{km}, & \text{if } i \neq j, \\ \delta_{pl}\,\delta_{km} + \delta_{pk}\,\delta_{lm} + \delta_{pm}\,\delta_{lk}, & \text{if } i = j. \end{cases}$$

with the Kronecker symbol $\delta_{ij} = 1$ if $i = j$, $0$ if $i \neq j$. Now we have:

$$\mathbb{E}_{p_h}\big[(\nabla_\theta \hat{I}_2)(\nabla_\theta \hat{I}_2)^\top\big] = \frac{1}{M^2}\Big[\sum_{i \neq j}\delta_{km}\delta pl(\nabla_\theta D_{km})(\nabla_\theta D_{pl})^T +$$
$$\sum_{i=j}(\delta_{pl}\,\delta_{km} + \delta_{pk}\,\delta_{lm} + \delta_{pm}\,\delta_{lk})(\nabla_\theta D_{km})(\nabla_\theta D_{pl})^T\Big], \quad (23)$$

which reads hence,

$$\mathbb{E}_{p_h}\big[(\nabla_\theta \hat{I}_2)(\nabla_\theta \hat{I}_2)^\top\big] = \frac{1}{M^2}\Big[\sum_{i \neq j}\sum_{p,l,k,m}(\nabla_\theta D_{kk})(\nabla_\theta D_{pp})^T +$$
$$\sum_{i=j}\sum_{p,l,k,m}(\delta_{pl}\,\delta_{km} + \delta_{pk}\,\delta_{lm} + \delta_{pm}\,\delta_{lk})(\nabla_\theta D_{km})(\nabla_\theta D_{pl})^T\Big]. \quad (24)$$

The first term equals:

$$(1 - \frac{1}{M})(\nabla_\theta Tr(D))(\nabla_\theta Tr(D))^T\big]. \quad (25)$$

As for the second term, we have:

$$\sum_{i=j}\sum_{p,l,k,m}(\delta_{pl}\,\delta_{km} + \delta_{pk}\,\delta_{lm} + \delta_{pm}\,\delta_{lk})(\nabla_\theta D_{km})(\nabla_\theta D_{pl})^T$$
$$= M\sum_{p,l,k,m}(\delta_{pl}\,\delta_{km} + \delta_{pk}\,\delta_{lm} + \delta_{pm}\,\delta_{lk})(\nabla_\theta D_{km})(\nabla_\theta D_{pl})^T\big], \quad (26)$$

which reads,

$$M\big(\sum_{p,k}(\nabla_\theta D_{kk})(\nabla_\theta D_{pp})^T + \sum_{p,l}(\nabla_\theta D_{pl})(\nabla_\theta D_{pl})^T + \sum_{p,l}(\nabla_\theta D_{lp})(\nabla_\theta D_{pl})^T)\big]$$
$$= M\big[(\nabla_\theta Tr(D))(\nabla_\theta Tr(D))^T + \sum_{p,l}(\nabla_\theta(D_{pl} + D_{lp}))(\nabla_\theta D_{pl})^T\big]. \quad (27)$$

The sum of the two terms (25) and (27) yields:

$$\mathbb{E}_{p_h}\big[(\nabla_\theta \hat{I}_2)(\nabla_\theta \hat{I}_2)^\top\big] = J(x)^2\Big[\big(\nabla_\theta Tr(D)\big)\big(\nabla_\theta Tr(D)\big)^T +$$
$$\frac{1}{M}\sum_{p,l}\big(\nabla_\theta(D_{pl} + D_{lp})\big)\big(\nabla_\theta D_{pl}\big)^T\Big], \quad (28)$$

which can be rewritten more compactly as:

$$\mathbb{E}_{p_h}\big[(\nabla \hat{I}_2)(\nabla \hat{I}_2)^\top\big] = \mathbb{E}_p\big[(\nabla_\theta I_2)(\nabla_\theta I_2)^\top\big] + \frac{1}{M}\Sigma_{MC}. \quad (29)$$

This complete the proof, and we have finally:

$$\Sigma = \underbrace{\mathbb{E}_p\Big[(\nabla I_1)(\nabla I_1)^\top + (\nabla I_1)(\nabla I_2)^\top + (\nabla I_2)(\nabla I_1)^\top + (\nabla I_2)(\nabla I_2)^\top\Big]}_{\Sigma_{\text{det}}} + \frac{1}{M}\,\Sigma_{\text{MC}},$$

where the second term corresponds to a Monte-Carlo approximation of the divergence term. Accordingly, the asymptotic variance of the WGM estimator is:

$$V = H^{-1}(\Sigma_{det} + \frac{1}{M}\Sigma_{MC})H^{-1}$$

When the divergence is computed exactly (i.e., in the deterministic case), the Monte Carlo term vanishes, and the variance reduces to $H^{-1}\Sigma_{det}H^{-1}$. Furthermore, if the gradient flow $v_\theta$ is represented as the gradient of a potential function, i.e., $v_\theta = \nabla NN_\theta$, then the Jacobian matrix $D$ is symmetric, and the Fisher information matrix $H$ is positive definite.

In this section we study the asymptotic behavior and variance associated to weak gradient estimation. We follow the classical notations :

$$\hat{\theta}_N = \arg\min_\theta \frac{1}{N}\sum_{i=1}^{N}\mathcal{L}(X_i, \theta) = \mathcal{L}_N(\theta)$$

Let:

- $\mathcal{L}_{MSE}$, The mean square error cost defined as:

$$L_{MSE}(x) = \mathbb{E}_p\big[\big(v_\theta(x) - \nabla J(x)\big)^T\big(v_\theta(x) - \nabla J(x)\big)\big].$$

- $\mathcal{L}_{WGM}$, The weak gradient matching cost :

$$L_{WGM}(x) = \mathbb{E}_p\Big[\big(v_\theta(x)\big)^T\big(v_\theta(x)\big) + J(x)\Big(div\big(v_\theta(x)\big) + \langle v_\theta(x), \nabla\log p(x)\rangle\Big)\Big]$$

In the next section we compare the direct gradient regression when using the cost $L_{MSE}$ or $L_{WGM}$ to point that the weak gradient approach naturally comes with higher variance as it rely not on direct observations of the targeted gradient.

### A.3.1 DIRECT GRADIENT MATCHING VS WGM ON A LINEAR CASE

The weak gradient matching (WGM) cost involves only scalar observations of a potential function. This lack of directional information, compared to a direct gradient-based approach, naturally leads to greater instability during training. We briefly illustrate the difference in variance between these two approaches using a simple linear example.

Since the WGM training cost relies solely on scalar evaluations of the potential function $J$, it is naturally associated with higher variance. This becomes clear in the following linear case.

Let $v_\theta(x) = \sum \theta_k f_k$ be a linear combination of fixed features functions $f_k$ and let $J(x) = \langle x, Ax\rangle$, be a quadratic form, where $A \in \mathbb{R}^{n\times n}$.

The gradient of the mean squared error loss $L_{MSE}$ with respect to $\theta_i$ is:

$$\frac{\partial L_{MSE}}{\partial\theta_i} = 2\mathbb{E}_p\Big[\sum_k \theta_k\langle f_i - \nabla J, f_k\rangle\Big]$$

In matrix form, this becomes:

$$\nabla_\theta L_{MSE} = \mathbb{E}_p\big[f(f^T\theta^* - \nabla J)\big] = 2M\theta + 2C$$

where $M_{i,j} \in \mathbb{R}^{n\times n} = \langle f_i, f_j\rangle$, and $C_i = \langle f_i, -\nabla J\rangle$. The optimal parameter $\theta^*$ statisfies $\nabla_\theta L_{dir} = 0$ yielding $\theta^* = M^{-1}C$.

A similar derivation for the WGM loss $L_{WGM}$ gives

$$\nabla_\theta L_{WGM} = 2M\theta + 2C_2$$

with $C_{2,i} = \mathbb{E}_p\Big[J(x)\Big(div\big(f_i(x)\big) + \langle f_i, \nabla log(p)\rangle\Big)\Big].$

Upon using an integration by parts under the distribution $p$, we find that $C_2 = C$. However, the second-order moments of the estimators $\hat{C}_2$ and $\hat{C}$ may differ. For linear functions $f_k(x) = B_k x$ and and a Gaussian distribution $p \sim \mathcal{N}(0, \Sigma)$, we have:

$$Var(\hat{C}_n) = \frac{1}{n} \sum Var_p\big[\langle B_k x, \nabla J(x) \rangle\big],$$

$$Var(\hat{C}_{2_n}) = \frac{1}{n} \sum \big[Var_p\big(J(x)(Tr(B_k) + \sum \theta_k (B_k x)^T \Sigma^{-1} x)\big]$$

If $J(x) = \langle x, Ax \rangle \varphi(x)$ (where $\varphi$ is a function of compact support ensuring $J_{|\partial\Omega} = 0$ on the domain boundary), then the terms inside the variance of $\hat{C}_2$ involve higher-degree polynomials than those in $\hat{C}$. As a result, we should generally expect the weak gradient estimator to exhibit a greater variance than the direct MSE estimator – which is unsurprising , given that it relies only on scalar observations of $J$, rather than the full vector $\nabla J$.

## A.4    Why surrogate modeling may not be always appropriate for estimating the gradient of a function

### A.4.1    A problem of orthogonality

In numerous domains, while the object we aim to estimate is the gradient $\nabla J$ of a scalar function $J$, the targeted object is estimated indirectly by first fitting a model $\hat{J}$ and then differentiating this surrogate. We justify here that even in the perfect scenario where an infinite number of samples $(x_i, J(x_i))$ is available, this strategy might be suboptimal.

Let $S = (\Phi_1, \ldots, \Phi_n)$ be a set of functions $\Phi_i : \mathbb{R}^n \to \mathbb{R}$. We denote by $P$ the projection operator onto the subspace spanned by $S$, i.e., $\mathrm{span}(S)$, and by $P_2$ the second projection operator which maps functions from $\mathbb{R}^n \to \mathbb{R}^n$ to $\mathrm{span}(\nabla S) = \mathrm{span}(\nabla \Phi_1, \ldots, \nabla \Phi_n)$. Generally, we have:

$$\nabla(P(f)) \neq P_2(\nabla f).$$

Indeed, using the decomposition $f = P(f) + f^\perp$, where $f^\perp$ denotes the component of $f$ orthogonal to $\mathrm{span}(S)$ (in the $L^2(p)$ sense), we have:

$$P_2(\nabla f) = P_2(\nabla(P(f) + f^\perp)) = P_2(\nabla P(f) + \nabla f^\perp) = \nabla P(f) + P_2(\nabla f^\perp).$$

The last equality holds because $\nabla P(f) \in \mathrm{span}(\nabla S)$, and $P_2$ acts as the identity on its image. Consequently, equality between the two estimators would require the second term $P_2(\nabla f^\perp)$ to vanish. This implies that for any $i \in \{1, \ldots, n\}$:

$$\langle \nabla \Phi_i, \nabla f^\perp \rangle_p = \int_\Omega \nabla \Phi_i(x) \cdot \nabla f^\perp(x)\, dp(x) = 0.$$

By Green's theorem (assuming boundary terms vanish), we can rewrite this inner product:

$$\int_\Omega \nabla \Phi_i \cdot \nabla f^\perp\, dp = - \int_\Omega \mathrm{div}(p \nabla \Phi_i) f^\perp\, dx$$

$$= - \int_\Omega (\Delta \Phi_i + \langle \nabla \log p, \nabla \Phi_i \rangle) f^\perp p(x)\, dx$$

$$= \langle -\mathcal{L}\Phi_i, f^\perp \rangle_p,$$

where $\mathcal{L}$ is the differential operator associated with the measure $p$, defined by $\mathcal{L}g = \Delta g + \langle \nabla g, \nabla \log p \rangle$. As $f^\perp \in \mathrm{span}(S)^\perp$, the orthogonality $\langle \mathcal{L}\Phi_i, f^\perp \rangle_p = 0$ holds specifically if $\mathcal{L}\Phi_i \in \mathrm{span}(S)$. This condition—that the approximation space be invariant under $\mathcal{L}$—is generally not met. For example, using the truncated Fourier basis $S = \{e^{\mathrm{i}2\pi kx/T}\}_k$ with a gaussian measure $p = \frac{1}{\sqrt{2\pi}} e^{-x^2/2}$ leads to a systematic bias as, for $g \in S$:

$$\mathcal{L}g = g'' + g'(log p)' = g'' - \frac{1}{2\sqrt{2\pi}} x g'$$

which can not be represented in $Span(S)$ due to the term $x g'$

This motivate the need a specific learning criterion to predict the gradient of a function rather than relying on a surrogate + autodiff approach.

## B  ALGORITHMS AND METHODOLOGICAL DISCUSSION

### B.1  KDE MARGINALIZATION TRICK FOR SCORE APPROXIMATION

A key challenge in this setup is that the sampling distribution $p(x)$ is a Kernel Density Estimate (KDE), which can be formulated as a Gaussian Mixture Model (GMM) over the entire dataset of $N$ corrupted images $\{z_i\}_{i=1}^N$:

$$p(x) = \frac{1}{N} \sum_{i=1}^N \mathcal{N}(x|z_i, \sigma^2 I) \tag{30}$$

The score of this mixture, $\nabla_x \log p(x)$, is computationally intractable as it requires summing over the entire dataset in the denominator for each point $x$.

To overcome this, we reformulate the learning objective. The ultimate goal is to minimize the Mean Squared Error between the estimated and true gradients under the distribution $p(x)$:

$$\mathcal{L}_{\mathrm{MSE}}(\theta) = \mathbb{E}_{x \sim p(x)} \left[ \|v_\theta(x) - \nabla J(x)\|^2 \right] \tag{31}$$

We can express the expectation over the mixture as an average of expectations over its individual components. By introducing a discrete latent variable $i \sim \mathrm{Uniform}(1, \dots, N)$ that selects a component center $z_i$, we can marginalize it out:

$$\mathcal{L}_{\mathrm{MSE}}(\theta) = \int p(x) \|v_\theta(x) - \nabla J(x)\|^2 dx \tag{32}$$

$$= \int \left( \frac{1}{N} \sum_{i=1}^N p(x|z_i) \right) \|v_\theta(x) - \nabla J(x)\|^2 dx \tag{33}$$

$$= \frac{1}{N} \sum_{i=1}^N \int p(x|z_i) \|v_\theta(x) - \nabla J(x)\|^2 dx \tag{34}$$

$$= \frac{1}{N} \sum_{i=1}^N \mathbb{E}_{x \sim p(x|z_i)} \left[ \|v_\theta(x) - \nabla J(x)\|^2 \right] \tag{35}$$

This reformulation shows that minimizing the global MSE is equivalent to minimizing the average of MSEs, where each term's expectation is taken over a single, tractable Gaussian component $p(x|z_i) = \mathcal{N}(x|z_i, \sigma^2 I)$.

Now, we can apply the Weak Gradient Matching identity (Proposition 3.1) to *each term* in the summation independently. This yields a practical loss function where the intractable score $\nabla_x \log p(x)$ is replaced by the simple, analytical score of the Gaussian component:

$$\nabla_x \log p(x|z_i) = -\frac{x - z_i}{\sigma^2} \tag{36}$$

The final objective to minimize is thus the expectation over the data points $z_i$ and the noise added to them:

$$\mathcal{L}_{\mathrm{WGM}}(\theta) = \mathbb{E}_{z_i \sim \mathrm{Data}} \left[ \mathbb{E}_{x \sim p(x|z_i)} \left[ \|v_\theta(x)\|^2 + 2J(x) \left( \mathrm{div}(v_\theta(x)) + \left\langle v_\theta(x), -\frac{x - z_i}{\sigma^2} \right\rangle \right) \right] \right] \tag{37}$$

This formulation is computationally efficient and matches the algorithm implemented: in each training step, we sample a mini-batch of corrupted data points $\{z_i\}$, draw a noisy sample $x$ for each $z_i$, and compute the loss using the simple analytical score of the corresponding Gaussian component.

### B.2  THE LEARNING SCHEME FOR AMORTIZED VARIATIONAL INFERENCE

The pseudo-code corresponding to the training procedure of the mimicking gradient scheme in the paper can be found below:

---

**Algorithm 3** Learning Procedure

---

**Require:** a scalar function $J$, a sampling distribution $p$, a learning rate $\eta$
1: Initialize parameters $\theta_0$
2: **for** $k = 1$ to $K$ **do**
3:     Integrate $V'_{\theta_k}(t) = -v_{\theta_k}(t)$ to obtain $V_{\theta_k}(t)$
4:     Extract $N$ points $V_i$
5:     Sample $N$ points $\epsilon_i$ from the distribution $p$
6:     Set $x_i = V_i + \epsilon_i$
7:     Evaluate $J(x_i)$
8:     Estimate the divergence of $v_{\theta_k}$ at $x_i$
9:     Compute the loss function $\widehat{\mathcal{L}}(\theta_k)$ based on $J(x_i)$ and $v_{\theta_k}(x_i)$
10:     Optimize $\widehat{\mathcal{L}}(\theta)$:
$$\theta_{k+1} \leftarrow \theta_k - \eta \nabla_\theta \widehat{\mathcal{L}}(\theta_k)$$
11: **end for**
12:
13: **return** $\theta_K$

---

### B.3    On the Parameterization: Conservative vs. Non-Conservative

In this work, we introduce a gradient field estimator that allows for parameterizing neural estimators in a non-conservative form. The use of a conservative form ($v_\theta = \nabla V_\theta$) is appealing for three main reasons:

- It allows for constraining the scalar potential $V_\theta$ itself, similar to surrogate modeling approaches.
- It provides a regularizing effect on the learned field.
- The Jacobian of the neural network becomes symmetric, which is advantageous for the Hutchinson estimation of the divergence.

A notable drawback, however, is that training such networks can involve third-order derivatives due to the terms required for the divergence computation, which can introduce computational and optimization challenges.

In our experiments, we show that for the problems and parameterizations considered, the use of a conservative form does not systematically lead to the best performance. Investigating the reasons for this observation is an interesting direction for future work. At least two hypotheses can be considered: (1) it is an optimization issue, or (2) there is an inductive bias in the neural architecture used (CNNs in this case) that favors the non-conservative approach. This finding is consistent with trends in other areas of machine learning. In numerous contemporary problems, including score matching for score-based diffusion techniques, the target object is also a gradient. From a practical perspective, most modern implementations do not enforce a conservative structure, opting instead for direct, non-conservative parameterizations.

### B.4    On the Navier-Stokes Experiment

The results of the Navier-Stokes data assimilation experiment are particularly noteworthy and merit further discussion. They also bring an other example of realistic scenario where it has been possible to apply WGM without the use of a generative model to modelize the data score $\nabla \log p_{data}$. We first recall the practical difference between amortized inference and optimization in this scenario.

#### B.4.1    On the Distinction Between Amortized Inference and Direct Optimization

A fundamental distinction must be drawn between direct optimization methods (such as CMA-ES (Hansen, 2023) ) and the Amortized Variational Inference approach proposed in this work (WGM). While both strategies aim to minimize a cost function $J$, they operate

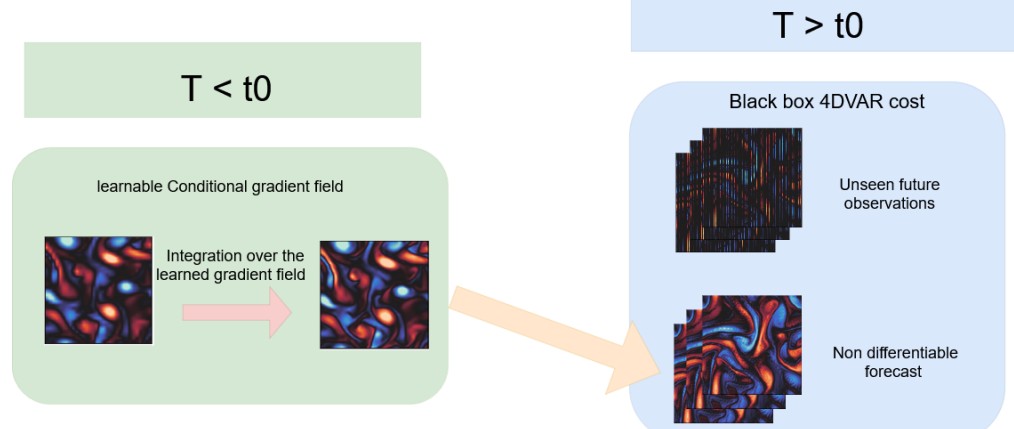

**Figure 6: Amortized Inference Strategy.** The learning problem is structured to reflect a forecasting scenario. The gradient estimator $v_\theta$ is conditioned solely on past information ($t \leq t_0$, green zone), such as low-resolution or noisy historical states. The cost function $J$, however, relies on future observations ($t > t_0$, blue zone) which are unavailable at inference time. While direct optimization methods cannot operate without access to these future values to evaluate the cost, WGM learns to predict the expected gradient direction offline, enabling operational forecasting.

under radically different constraints regarding data accessibility and causality. While SPSA is mainly used for optimization, in this paper we used it as an alternative criterion for learning a reusable gradient field.

In our Navier-Stokes experiment, we explicitly structure the learning problem to reflect an operational forecasting scenario. As illustrated in Figure 6, the dataset is temporally split into two disjoint sets relative to a reference time $t_0$:

- **Conditional Inputs ($t \leq t_0$):** These represent the information available at inference time (e.g., past noisy observations, low-resolution priors). This is the *only* information fed to the gradient estimator $v_\theta$.

- **Evaluation Targets ($t > t_0$):** These are the "future" high-resolution observations used solely to compute the 4D-Var cost function $J(x_0) = \sum_{t=t_0}^{T} \|\mathcal{H}(\mathcal{M}(x_0, t)) - y_t\|^2$.

**The Operational Paradox of Optimization.** In a real-world deployment (test time), an agent must infer the optimal initial condition $x_0^*$ to minimize the forecast error *before* the future observations $y_{t>t_0}$ occur. Standard optimization methods (black-box or adjoint-based) are iterative: they require evaluating the cost $J(x)$ (and its gradient) at the current candidate solution to propose a new one. However, since $J$ depends on future data $y_{t>t_0}$ which is unavailable at $t_0$, **direct optimization is strictly impossible in this setting**. One cannot calculate the gradient of a cost function that cannot yet be evaluated.

**WGM as a conditional gradient field predictor.** Conversely, the Weak Gradient Matching framework is used here for *amortized* inference: during the offline training phase, the model has access to a historical database containing both past inputs and their corresponding future outcomes. The estimator $v_\theta$ learns to predict a conditional gradient field associated to the future cost. Mathematically, WGM approximates the conditional expectation of the gradient:

$$v_\theta(x_{\text{past}}) \approx \mathbb{E}_{y_{\text{future}}|x_{\text{past}}} \left[ \nabla_{x_0} J(x_0, y_{\text{future}}) \right] \tag{38}$$

Once trained, the model acts as an oracle: given only the past ($t \leq t_0$), it predicts the direction most likely to minimize the error against the unseen future. This capability allows WGM to perform "optimization" in real-time without ever needing to evaluate the cost function itself, effectively bypassing the causality constraint that binds traditional solvers.

### B.4.2 Learning from KDE with Isotropic Noise: a surprising result

A surprising and powerful outcome of this work is the demonstration that WGM can solve a complex inverse problem, which can be interpreted as a **physically-constrained super-resolution task**. The model successfully learns to infer the high-frequency details of an initial condition required to minimize a future forecast error. Remarkably, it achieves this using a training procedure that relies solely on sampling from a simple, high-variance isotropic Gaussian distribution ($p = \mathcal{N}(0, I)$). This shows that with a suitable learning objective, structured and physically consistent information can be extracted from unstructured exploration, a result of significant practical importance for scientific machine learning.

### B.4.3 On the Failure of Surrogate and Finite-Difference Methods

The success of WGM is further highlighted when contrasted with the failure of the baseline methods in this challenging setting.

- **Surrogate Modeling:** Consistent with our findings in other experiments (MNIST), the surrogate-based approach failed to yield meaningful descent directions. Although the surrogate network could be trained to approximate the scalar 4DVAR cost, its gradients, obtained via automatic differentiation, were not reliable enough to effectively minimize the true cost function. This result strongly reinforces the central thesis of our work: for complex systems, directly estimating the gradient field is a more robust strategy than differentiating an imperfect surrogate.

- **Random Finite-Differences (SPSA):** The finite-difference method also failed. We hypothesize that this is due to its fundamental reliance on small, localized perturbations of the input state. In a dissipative physical system like the one governed by the Navier-Stokes equations, the effects of such minor perturbations to the initial condition are often quickly dampened or smoothed out by the forward simulation. Consequently, the resulting difference in the cost function, $J(x + \epsilon v) - J(x - \epsilon v)$, can become vanishingly small, often falling below the limits of numerical precision and thus failing to provide a useful gradient signal for learning.

In contrast, the WGM approach proved effective precisely because it utilizes a high-variance sampling distribution. The standard deviation of this "exploration noise" was intentionally set to be of a similar order of magnitude (we observed better results with high variance exploraiton) as the standard deviation of the true high-frequency residual we aimed to model. This suggests that for WGM to succeed in such scenarios, the sampling distribution must be sufficiently broad to explore variations that are as significant as the target function's features.

### B.5 Computational cost.

In this section we discuss about the computational cost associated to different gradient method estimations. The choice of a method over an other is a compromise between the sampling budget / $J$-evaluation and the neural network training budget.

Table 5 summarizes the per-step training complexity. All methods require $\mathcal{O}(N\,C_J)$ simulator calls per batch, except SPSA which doubles this due to two-point perturbations. WGM matches Surrogate-AD on simulator calls but adds $M$ Hutchinson directions to approximate $\mathrm{div}(v_\theta)$: with $v_\theta = \nabla\Phi_\theta$, each direction is a Jacobian–vector product implemented as a backward-only pass, yielding $\mathcal{O}(N\,(C_{\mathrm{net}} + M\,C_{\mathrm{bwd}}))$ network cost. In regimes where the simulator dominates ($C_J \gg C_{\mathrm{net}}$) and with small $M$ (we use $M \in \{1, 5\}$; $M{=}1$ by default), WGM is competitive in wall-clock while providing lower-variance gradients than SPSA and more faithful descent directions than Surrogate-AD.

### B.6 General discussion

The proposed Weak Gradient Matching (WGM) framework introduces a novel and practical principle for estimating gradients field from scalar evaluations alone. Our experiments

| Cost Component | SPSA | Surrogate-AD | WGM |
|---|---|---|---|
| $J(x)$ Evaluations | $\mathcal{O}(2N \cdot C_J)$ | $\mathcal{O}(N \cdot C_J)$ | $\mathcal{O}(N \cdot C_J)$ |
| Network Passes (fwd+bwd) | $\mathcal{O}(N \cdot C_{net})$ | $\mathcal{O}(N \cdot C_{net})$ | $\mathcal{O}(N \cdot (C_{net} + M \cdot C_{bwd}))$ |

**Table 5:** Per-step training cost for a batch of size $N$. $C_J$: cost of one black-box evaluation $J(x)$. $C_{\text{net}}$: cost of a single forward+backward pass through the estimator network $\Phi_\theta$ (when $v_\theta = \nabla\Phi_\theta$). $C_{\text{bwd}}$: cost of a backward-only pass (used by Hutchinson). $M$: number of Hutchinson vectors per sample.

demonstrate that it offers a robust alternative to surrogate modeling and finite-difference methods, especially in noisy and high-dimensional settings where traditional techniques often falter. A key insight is that WGM directly optimizes an objective tied to the gradient field, avoiding the pitfalls of differentiating an imperfect surrogate model. The method's computational efficiency, relying on a single function call and a stochastic trace estimator, makes it scalable to complex scientific problems like the 2D Navier-Stokes assimilation task presented. Furthermore, the inherent connection to Sobolev-type objectives when using a potential-based estimator provides a principled avenue for developing gradient-aware regularizers. Despite its promise, our work opens several important questions and highlights areas for future research. The most significant limitation is the reliance on a known score function of the sampling distribution, $\nabla \log p(\mathbf{x})$. As shown in numerical experiments, this limitation can be tempered: The experiment on MNIST show that there exists scenarios where it is possible to learn from a bad model of the data while on the Navier-Stokes experiment we show that it was possible to learn a convincing Physically constrained "super-resolution" model without the use of automatic differentiation solely from gaussian priors.

## C  COMPLEMENTARY EXPERIMENTS & ABLATION STUDY

### C.1  SENSITIVITY ANALYSIS

#### C.1.1  SENSITIVITY TO THE NUMBER OF HUTCHINSON SAMPLES

A key component of the WGM loss is the approximation of the divergence term $\text{div}(v_\theta)$ using the Hutchinson trace estimator. This estimator relies on $M$ random projections onto Gaussian vectors. To assess the trade-off between computational cost and estimation accuracy, we performed an ablation study on the MNIST super-resolution task by varying $M \in \{1, 4, 8, 12, 16\}$.

Figure 7 reports the Mean Squared Error (MSE) and Cosine Similarity between the estimated gradient field and the ground truth.

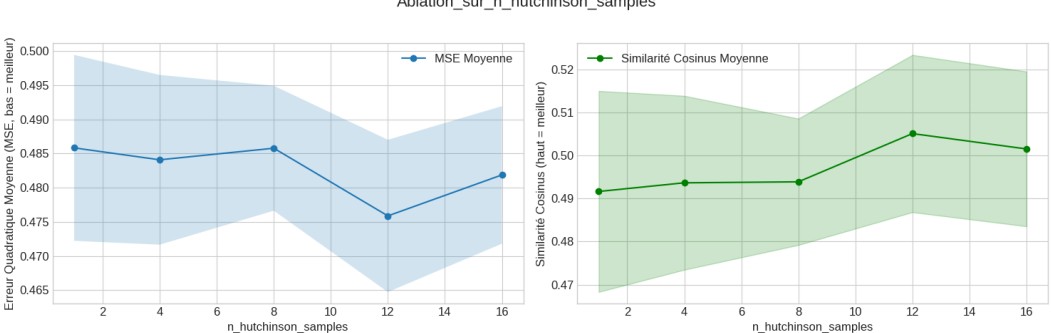

**Figure 7: Impact of Hutchinson samples ($M$).** Left: Mean Squared Error (lower is better). Right: Cosine Similarity (higher is better). Shaded areas represent the standard deviation over multiple runs. The performance remains stable even when $M$ is small.

**Analysis.** We observe only a marginal improvement in Cosine Similarity when increasing $M$ from 1 to 12, while the computational cost of the divergence term scales linearly with $M$. In the experiment on Navier Stokes we use $M = 1$ which we consider as a good efficiency-accuracy trade-off for high-dimensional problems.

### C.1.2 SENSITIVITY TO THE CONTROL VARIATE

We empirically evaluate the impact of the control variate baseline $C$ on the stability and accuracy of the WGM estimator. We recall that the loss involves a term proportional to $(J(x) - C)$. While the expectation of the gradient remains unbiased for any constant $C$, the variance of the estimator is highly sensitive to this choice.

Let $C = c\mu$ with $\mu = \mathbb{E}_i[J(x)]$.

We performed an ablation on the MNIST super-resolution task by varying the control variate multiplier $c$. Figure 8 reports the Mean Squared Error (MSE) and Cosine Similarity for different values of $c$.

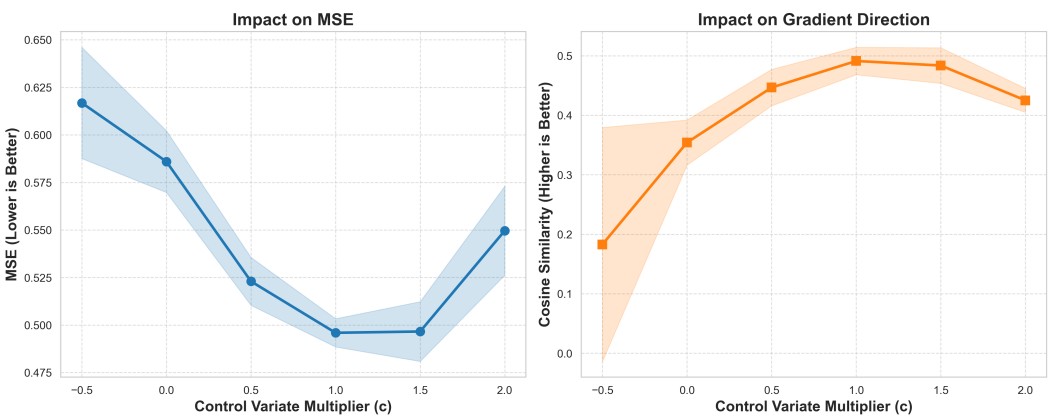

**Figure 8: Impact of the Control Variate ($c$).** Left: Mean Squared Error (lower is better). Right: Cosine Similarity (higher is better). The x-axis represents the multiplier applied to the baseline subtraction (where $c = 1$ corresponds to centering $J(x)$). Shaded areas represent the standard deviation. The results confirm that a well-chosen control variate ($c \approx 1$) significantly improves gradient quality and reduces estimation variance compared to no baseline ($c = 0$).

**Analysis.** The results empirically confirm our theoretical derivation. We observe a clear optimum around $c = 1.0$, which minimizes the MSE and maximizes the Cosine Similarity with the ground truth gradient. Crucially, the shaded regions (standard deviation over runs) demonstrate that using an appropriate control variate significantly reduces the variance of the training process. In contrast, omitting the control variate ($c = 0$) or using an incorrect sign ($c < 0$) leads to higher instability and poorer convergence. Consequently, we adopt centered scalar evaluations ($c \approx \mathbb{E}[J]$) as the default strategy for all experiments.

### C.2 COMPLEMENTARY EXPERIMENTS

#### C.2.1 EXAMPLE: LINEAR PARAMETERIZATIONS WITH GAUSSIAN KERNELS

As the weak gradient matching cost is convex and involves linear operators, it is particularly well-suited for estimators that are linear in their parameters, $v_\theta(x) = \sum_i \theta_i \phi_i(x)$. This class of models includes finite element methods (Reddy, 1993), polynomial regression (**?**), and Radial Basis Function (RBF) networks, such as those using Gaussian kernels. For such parameterizations, the optimization problem can be solved optimally and the solution given explicitly. Figure 9 compares different strategies using an identical Gaussian kernel setup. The results show that minimizing different objectives can lead to distinct gradient

approximations. While surrogate models aim for functional fidelity, their derivatives are not always accurate. In contrast, WGM directly optimizes an objective tied to the gradient error, which can yield a more faithful gradient field estimation, as reflected by the mean squared error and cosine similarity metrics.

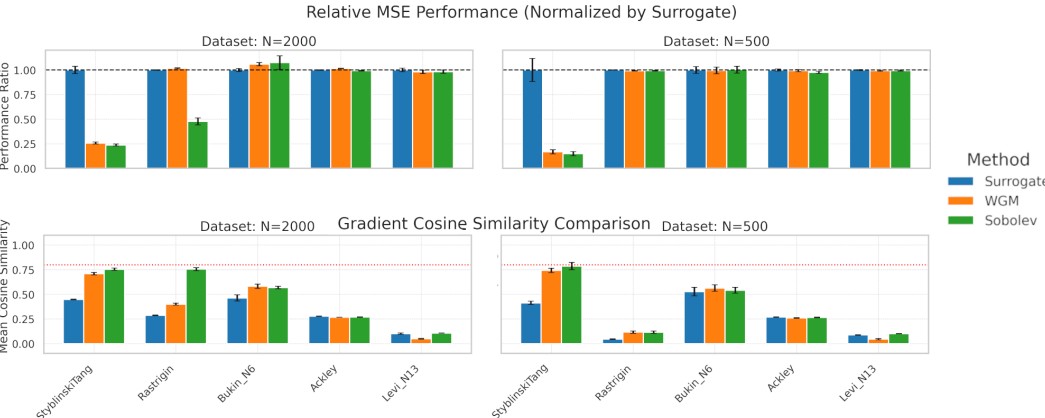

**Figure 9:** Top: comparison of normalized mean square error ($\frac{\text{MSE(method)}}{\text{MSE(Surrogate)}}$) for different methods (lower is better). Bottom: cosine similarity between the learned and target gradient fields (Higher is better).

### C.3 GEOSTROPHIC CURRENT ESTIMATION FROM SEA SURFACE HEIGHT OBSERVATIONS

In this experiment, we apply the WGM framework to a real-world inverse problem in oceanography: the estimation of **geostrophic currents** from sparse satellite altimetry data. In physical oceanography, surface currents are proportional to the spatial gradient of the Sea Surface Height (SSH), making the accurate recovery of the gradient field $\nabla$SSH as critical as the scalar field itself.

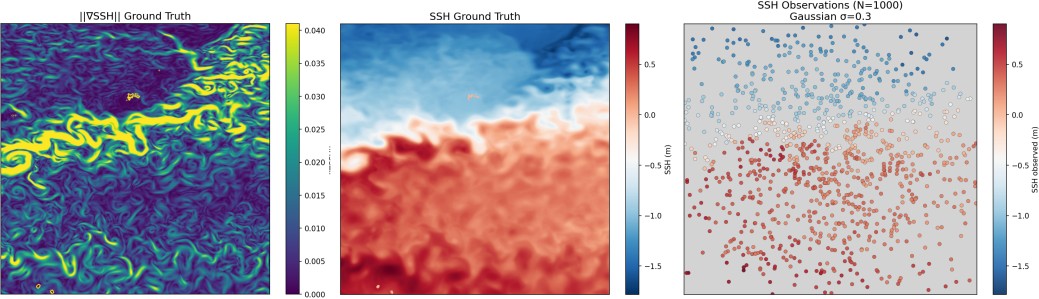

**Figure 10: Geostrophic Current Estimation from Sparse Altimetry Data.** Visualization of the inverse problem. **Left:** The ground truth gradient norm $\|\nabla\text{SSH}\|$, which is physically proportional to the magnitude of geostrophic currents. This contains the high-frequency structures we aim to recover. **Center:** The ground truth scalar Sea Surface Height (SSH) field. **Right:** The actual input available to the model: a sparse set of $N = 1000$ scalar observations sampled from a Gaussian distribution ($\sigma = 0.3$). Standard regression methods struggle to infer the complex gradient structures (left) solely from these sparse points (right).

**Experimental Setup.** As illustrated in Figure 10, the task consists of reconstructing the SSH field from sparse, noisy scalar observations. The field is parameterized using a Fourier basis with $M = 15$ modes, resulting in an input dimension $d = (2M + 1)^2 = 961$. The dataset consists of $N_{\text{train}} = 4000$ and $N_{\text{test}} = 1000$ samples.

We compare a standard Surrogate model trained with MSE loss against a model trained with a **Weak Sobolev** objective: $\mathcal{L}(\theta) = \mathcal{L}_{\text{MSE}}(\theta) + \lambda\mathcal{L}_{\text{WGM}}(\theta)$, with $\lambda = 0.5$. This effectively

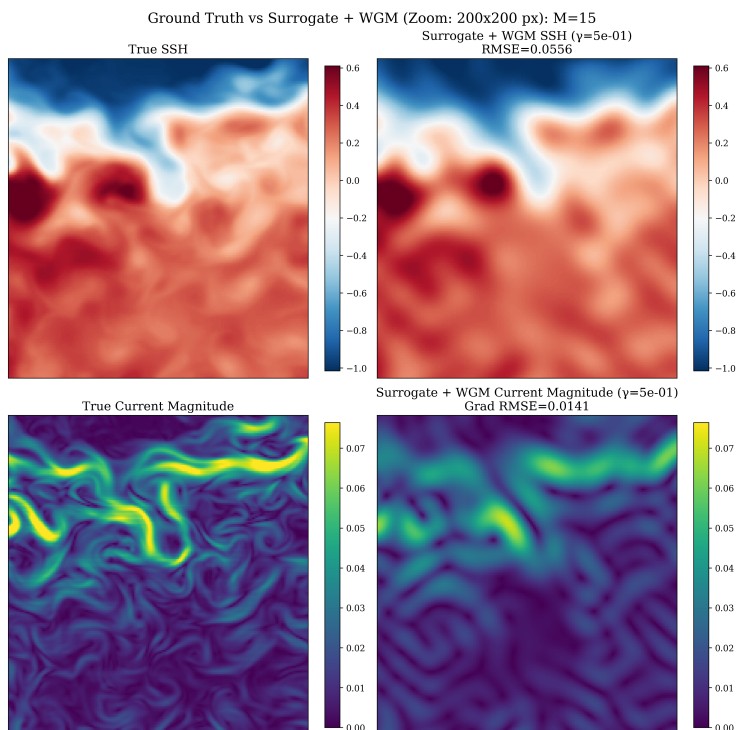

**Figure 11:** Top left: ground-truth ssh; top right: reconstructed SSH. Bottom left: magnitude of ground-truth SSH gradients. Bottom right: reconstructed gradient fields estimated by minimization of the Sobolev-WGM norm over fourier feature space.

enforces a prior on the derivatives without requiring ground-truth gradient data. In this context the optimization problems associated to each regression problem are solved globally as we optimize convex cost functions. Interestingly as shown in table 6 and as discussed in A.4.1 there is a slight trade off when solving this minimization issue as minimizing a sobolev norm leads in a slight sacrifice on train set metrics compared to a surrogate approach, on the test set therfore Sobolev norm act as a regularizer and leads to drastic improvement in terms of reconstruction metrics this might be due to outlier on area with poor sampling.

**Table 6:** Comparison of Surrogate vs. Weak Sobolev Training ($\lambda = 0.5$) on the SSH reconstruction task (mean $\pm$ std over 5 seeds). WGM acts as a powerful regularizer, significantly reducing test error for both the scalar field (SSH) and the physical quantity of interest (Currents/Gradients).

| Method | Train Set RMSE | | Test Set RMSE | |
|---|---|---|---|---|
| | Gradient ($\nabla J$) | Scalar ($J$) | Gradient ($\nabla J$) | Scalar ($J$) |
| Surrogate ($\lambda = 0$) | $0.033_{\pm 0.004}$ | $\mathbf{0.061}_{\pm \mathbf{0.004}}$ | $0.040_{\pm 0.016}$ | $0.265_{\pm 0.225}$ |
| **Weak Sobolev** ($\lambda = 0.5$) | $\mathbf{0.029}_{\pm \mathbf{0.003}}$ | $0.062_{\pm 0.004}$ | $\mathbf{0.030}_{\pm \mathbf{0.006}}$ | $\mathbf{0.153}_{\pm \mathbf{0.027}}$ |
| *Improvement* | *-11.4%* | *+0.4%* | *-23.7%* | *-42.2%* |

### C.3.1 COMPLEMENTARY EXPERIMENTS WITH NEURAL NETWORKS

This section complementary numerical experiments on toy target functions involving neural network parameterizations. We apply our proposed framework to some analytical functions used in (Czarnecki et al., 2017). While the WGM framework allow us to train arbitrary neural network, a gradient parameterization $v_\theta = \nabla f_\theta$ allows us to compare the implicitly learned surfaces as shown in figure 12. Quantitative metrics are reported on table 7.

1566
1567
1568
1569
1570
1571
1572
1573
1574
1575
1576
1577
1578
1579
1580
1581
1582
1583
1584
1585
1586
1587
1588
1589

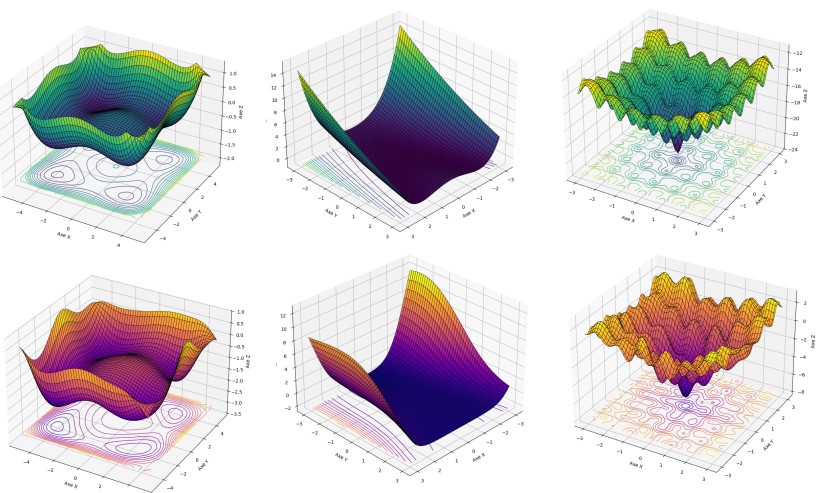

**Figure 12:** Top: ground truth surfaces: Stiblinsky function (left), Rosenbrock (middle), Anckley (right). Bottom: surface implicitly learned by multi-layers perceptron parameterized as the gradient of a scalar function and trained by WGM.

1590
1591
1592
1593
1594
1595
1596
1597
1598
1599
1600
1601
1602
1603
1604
1605
1606
1607
1608
1609
1610
1611
1612
1613
1614
1615
1616
1617
1618
1619

| XP | wgm | surrogate + AD | Sobolev regularization (mixed) |
|---|---|---|---|
| Rosenbrock | **9.0478 $\pm$ 1.9643** | 10.276 $\pm$ 1.9210 | 10.1429 $\pm$ 1.8125 |
| Styblinski | **0.0071 $\pm$ 0.00056** | 0.0086 $\pm$ 0.0002 | 0.0081 $\pm$ 0.00051 |
| Ackley | 0.0843 $\pm$ 0.0055 | **0.0819 $\pm$ 0.0013** | 0.0842 $\pm$ 0.002 |

**Table 7:** Results in terms of RMSE w.r.t ground-truth functions gradients averaged over 3 experiments.

### C.3.2 Offline Black-Box Adversarial Attacks via Learned Gradient Fields

In this section, we demonstrate a practical application of the proposed Weak Gradient Matching (WGM) framework: generating adversarial examples for a black-box classifier in an **offline** setting. Unlike standard zeroth-order attacks (e.g., SPSA, ZO-SignSGD) which require querying the target model thousands of times *during* the generation of a single adversarial example, our approach is **amortized**: we train a gradient estimator $v_\theta$ once, and can subsequently generate adversarial perturbations for any input without further queries to the target model.

Experimental Setup

**Target Model & Objective:** We target a pre-trained CNN classifier on MNIST trained to distinguish between *Even* and *Odd* digits. Let $f(x)$ be the classifier's logit output for the "Odd" class. The objective is to minimize this output (gradient descent) to flip the prediction to "Even". The ground-truth gradient $\nabla_x f(x)$ is accessible only for evaluation purposes (oracle).

**Gradient Estimators:** We compare three methods trained to estimate $\nabla_x f(x)$ using the *Mimicking Gradient Descent* scheme introduced in Section 3.4.1:

- **WGM-C / WGM-NC:** Our proposed method using Conservative ($\nabla V_\theta$) and Non-Conservative ($v_\theta$) parameterizations.
- **SPSA:** A baseline where the gradient field is trained to regress finite-difference estimates.
- **Surrogate:** A scalar proxy model $J_\theta \approx f$ differentiated via automatic differentiation.

**Attack Protocol:** We perform an untargeted attack by simply running Gradient Descent on the learned gradient flow., we use the raw output of the estimator $v_\theta$ to update the input iteratively: $x_{t+1} = x_t - \alpha v_\theta(x_t) - \nabla_x R(x)$. with $R$ a regularization term (l1 or l2 norm). We use an attack budget of 200 steps with a learning rate $\alpha = 0.3$.

Quantitative Results

Table 8 summarizes the attack performance. The **Success Rate** measures the percentage of images successfully misclassified (flipped from Odd to Even) after perturbation. **Cos Sim** indicates the cosine similarity between the estimated gradient and the true white-box gradient.

**Table 8:** Attack performance and Gradient Quality on MNIST (Even vs Odd). We report the top-performing configurations for each estimator type.

| Method | steps K during training | Success Rate | Mean $L_2$ | Cos Sim |
|---|---|---|---|---|
| **WGM-C (Ours)** | **7** | **41.0%** | 14.23 | 0.381 |
| WGM-C (Ours) | 5 | 39.0% | 14.21 | 0.358 |
| SPSA (Baseline) | 7 | 37.0% | 14.18 | **0.425** |
| WGM-NC (Ours) | 7 | 34.0% | 14.24 | 0.280 |
| WGM-C (Ours) | 0 | 28.0% | 14.15 | 0.235 |
| Surrogate (Baseline) | 7 | 20.0% | 14.19 | 0.177 |

**Key Findings:**

1. **Peak Performance:** The conservative WGM estimator (WGM-C) with $K = 7$ achieves the highest success rate (**41%**), outperforming the SPSA baseline (37%) and significantly surpassing the Surrogate approach (20%).
2. **Impact of Trajectory Learning:** There is a consistent improvement in attack success as $K$ increases. For WGM-C, training on trajectory points ($K = 7$) versus random sampling ($K = 0$) improves the success rate by +13 percentage points. This confirms that the *Mimicking Gradient Descent* scheme helps the estimator generalize to the states visited during the attack optimization.

Qualitative Analysis

Figure 13 illustrates adversarial examples generated by WGM-C ($K = 7$). The perturbations (right column) exhibit coherent structural features that align with the strokes of the digits. Crucially, these examples were generated **offline**. Once trained, the WGM network generates these attacks efficiently (one forward pass per step), whereas a standard zeroth-order attack would have required thousands of queries to the oracle classifier for *each* image.

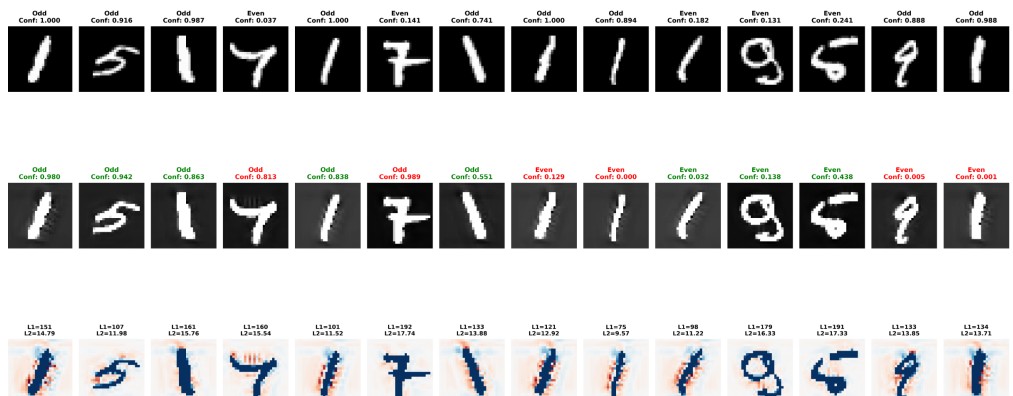

**Figure 13: Qualitative results of offline adversarial attacks.** Top: Original images (Odd digits) and output values of the trained classifier. Middle: Adversarial examples generated using WGM-C ($K = 7$).Bottom: performed perturbations.

## D  GIBBS FREE ENERGY ASSOCIATED TO 4DVAR ALGORITHMS

The construction of statistical models in physics often involves an unnormalized probability density function of the form $p(x) \propto e^{-E(x)}$, where $E(x)$ is an energy function that defines the distribution up to a normalization constant. This formulation is widely used to describe equilibrium states in various physical systems. Examples of Gibbs energy models include the Ising model (Cipra, 1987), which characterizes spin interactions in ferromagnetic materials, and the Boltzmann distribution in statistical mechanics, which governs the energy distribution of particles in a system. One of the most common methods for sampling energy distributions is the Markov Chain Monte Carlo (MCMC) algorithm (Andrieu et al., 2003), with Langevin sampling being a widely used approach. Langevin sampling can be interpreted as a noisy gradient descent on the Gibbs energy and is given by:

$$dx_t = -\nabla E(x_t)dt + \sqrt{2\beta^{-1}}d\xi(t), \tag{39}$$

where $d\xi(t)$ is a Gaussian noise term with independent increments. The Gibbs energy defines the stationary distribution of the Fokker-Planck equation (Markowich & Villani, 2000)associated with the Langevin dynamics. As a result, access to its gradient enables efficient sampling. In the field of generative modeling Energy-based models (EBMs) (Du & Mordatch, 2019) leverage this formulation by directly modeling the energy function $E(x)$ rather than the probability distribution itself, making them particularly useful for tasks such as generative modeling, denoising, and probabilistic inference.

### D.1  4DVAR GIBBS ENERGY

As described earlier the strong constraint 4DVAR algorithm corresponds to the following minimization problem:

$$x_0^* = \arg\min_{x_0} J_{SC}, J_{SC} = \sum_{i=0}^{T} \|\mathcal{H}_i(\mathcal{M}(x_0, t_i) - y_i\|_{\Sigma_i}^2.$$

Providing accurate probabilistic models to represent uncertainty around a state estimate is one of the main challenges in data assimilation. In the case of the 4DVAR algorithm, a

possible approach to constructing such a probabilistic model is to find the distribution that minimizes the expected forecast error for a fixed level of uncertainty, quantified in terms of differential entropy: $\mathcal{H}(p) = \int_{\Omega} p(x) \log p(x) \, dx$.

For a level of fixed uncertainty $\mathcal{H} = K$

$$p^* = \min_{p \in L^1(\Omega)} \mathbb{E}_{p(x)} \mathcal{J}_{SC}(x), \tag{40}$$

$$\text{subject to} \quad H(p) = K, \int p(z) dz = 1.$$

The solution to this optimization problem corresponds to a Gibbs distribution that depends on the parameter $K$ as stated in the following lemma:

**Lemma D.1.** *The optimal distribution, solution of problem (40), corresponds to the Gibbs energy, given by:*

$$p(x) = \frac{1}{Z\big(\lambda(K)\big)} \exp\left(-\frac{\mathcal{J}_{SC}(x)}{\lambda(K)}\right),$$

*where $\lambda(K)$ is a Lagrange parameters which depends on $K$ and $Z\big(\lambda(K)\big)$ is the partition function that normalizes the probability distribution, and $\lambda(K)$ is a parameter that depends on the entropy constraint $K$.*

**Proof:** The Lagrangian associated to the convex constrained optimization issue 40 is given by:

$$\mathcal{L}(p, \lambda_0, \lambda) = \langle p, J \rangle + \lambda_0\big(1 - \langle p, 1 \rangle\big) + \lambda\big(K - \langle p, \ln(p) \rangle\big)$$

With $J(z) = \mathbb{E}_p(J(z))$. The Karush-Kuhn-Tucker conditions states that an optimal $p^*$ verifies :

$$\frac{\partial L(p^*, \lambda, \lambda_0)}{\partial p} = \langle ., J \rangle - \lambda_0 \langle ., 1 \rangle + \lambda \langle ., \ln(p) + 1 \rangle = 0.$$

By scalar product with $\delta_x$:

$$J(x) + \lambda_0 + \lambda\Big(\ln\big(p(x)\big) + 1\Big) = 0,$$

which leads to:

$$p(x) = \exp\Big[-\frac{J(x) + \lambda_0}{\lambda} + 1\Big] = Z(\lambda_0, \lambda) \exp\big(-\lambda^{-1} J(x)\big)$$

With $Z(\lambda_0, \lambda) = \exp\big(1 - \frac{\lambda_0}{\lambda}\big)$. As $p$ is a probability density, the partition function $Z(\lambda_0, \lambda)$ verifies:

$$Z(\lambda_0, \lambda) = \int \exp\big(-\frac{J(x)}{\lambda(K)}\big) dx$$

The Lagrange multiplier $\lambda(K)$ depends on the entropy of the convex constrained problem and verifies:

$$\mathcal{H}(p) = K = \int p(x) \ln\big(p(x)\big) dx.$$

From a sampling perspective, as stated in section D, the knowledge of $\nabla_x \mathcal{J}(x, y)$ is sufficient to sample under $p$ through Langevin dynamics.

### D.2 LORENZ 63 4DVAR GIBBS ENERGY

The Lorenz 63 illustrates a typical problem in assimilation: the absence of ground truth. In practice, we only have access to partial and noisy data from the system as illustrated in the figure. It is governed by the following set of differential equations:

$$\begin{cases} \frac{dx}{dt} = \sigma(y - x), \\ \frac{dy}{dt} = x(\rho - z) - y, \\ \frac{dz}{dt} = xy - \beta z, \end{cases} \tag{41}$$

where $\sigma$, $\rho$, and $\beta$ are parameters controlling the system's evolution. The system exhibits chaotic behavior for certain values of these parameters, meaning that small variations in initial conditions lead to exponentially diverging trajectories over time. In the absence of ground truth, most data-based work in assimilation relies on simulated data, generalization to real data with unknown distribution is challenging. The ability to learn hybrid systems directly from partial data, as implemented in 4D-Var-type experiments, is thus crucial..

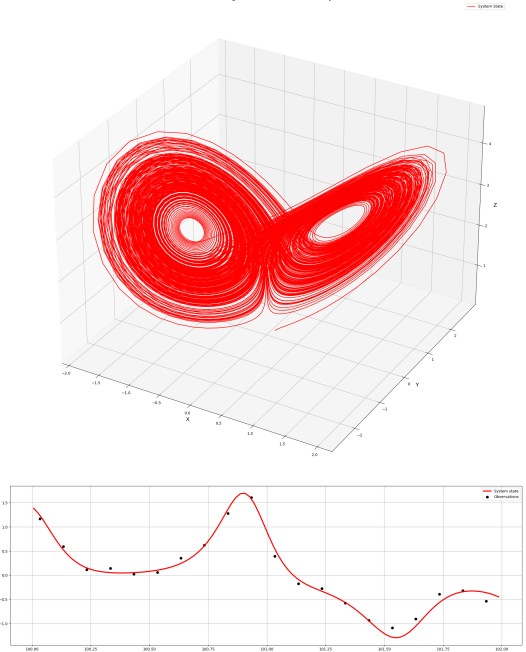

**Figure 14:** Top: This figure displays the ground-truth states of the Lorenz system. These states represent the actual trajectory of the system but are never directly observed in practice. In many real-world applications, we do not have direct access to the complete system state due to limitations in measurement capabilities. Bottom: the black dots correspond to the observed data, which are partial and noisy measurements. In our case, only the first component $x$ of the system is observed, and this observation is in addition corrupted by additive Gaussian noise. This reflects a common scenario in geophysical applications, where only some variables can be measured with limited accuracy.

In order to show the ability of the proposed framework to address Gibbs energy estimate in the context of data assimilation we conduct a complementary experiment on the Lorenz 63 system. We suppose that we only observe the first component through a noisy observational model:

$$\mathcal{H}(x, y, z) = x + \eta,$$

and where $\eta$ is a random white noise with $\sigma = 0.1$. We extract couples of time series $P_i, F_i$ containing both $T = 7$ one-dimensional noisy observations of the first variable $x$. The time series $P_i$ corresponds to past observation available up to a time $t_0$ while the observations $F_i$ stands for future observations. It is supposed that an estimate of the system state $\hat{X}_0 = f(C_i)$ is provided. For uncertainty modeling purpose, as discussed in D.1, we aim to predict gradients of the 4DVAR cost associated with this inverse problem:

$$J(X_0, Y_i) = \sum_{k=1}^{T_f} \|\mathcal{H}(\mathcal{M}(X_0 + k\Delta t)) - Y_i\|^2 \qquad (42)$$

We choose here to parameterize our neural network architecture as the gradient of a scalar function:

$$v_\theta(X_0, C_i) = \nabla_{X_0} \Phi_\theta(X_0, C_i)$$

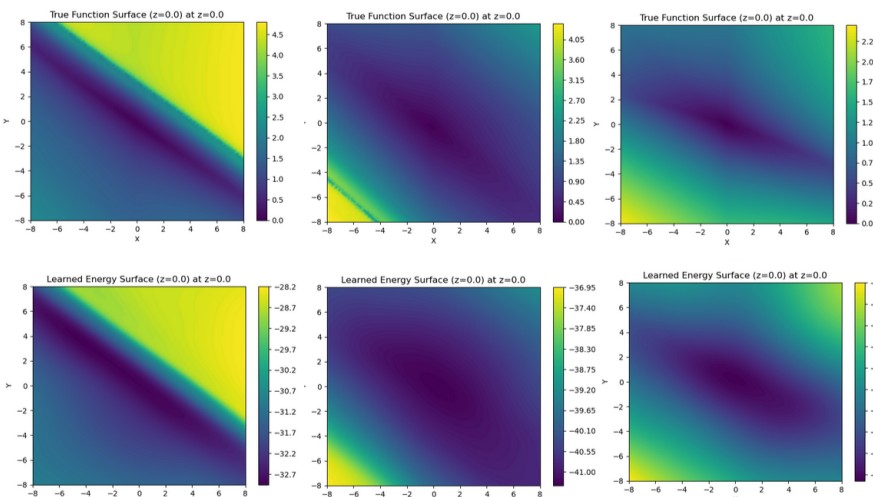

**Figure 15:** A comparison between ground-truth and predicted 4DVAR energies on the Lorenz 63. On top, the ground-truth energy associated to three different initial condition, on bottom the predicted surface.

This choice allows us to compare the conditional estimate $v_\theta(C_i)$ the learned expectation associated to the 4DVAR gradient above, with the ground-truth gradient associated to a time series $Y_i$. Results are shown in figure 15.

## E  Implementations details

### E.1  Anonymous repo

https://anonymous.4open.science/r/WGM-850D/

### E.2  Experiment on Gibbs measures

#### E.2.1  Implementation Details for Gibbs Energy Experiments

This section details the experimental setup for estimating the gradients of Gibbs energy functions, as presented in Table 2 of the main paper. The objective is to provide sufficient information for full reproducibility.

Target Gibbs Energy Functions The experiments are conducted on two classical models from statistical physics defined on a 2D lattice. The input $x$ is a tensor of shape `(1, 32, 32)`, representing the state on the grid.

1. **Ising Model:** The Ising model describes the interaction of magnetic spins. The energy (Hamiltonian) is given by:

$$E(x) = -J \sum_{\langle i,j \rangle} s_i s_j - h \sum_i s_i \tag{43}$$

where $s_i = \tanh(x_i)$ represents the spin at site $i$, $\langle i,j \rangle$ denotes summation over nearest-neighbor pairs, $J$ is the coupling constant, and $h$ is an external magnetic field. The nearest-neighbor interactions are implemented efficiently using a 2D convolution. We test three configurations:

   - **Ferromagnetic:** `J=1.0`, with `h=0.0` and `h=0.1`.
   - **Antiferromagnetic:** `J=-1.0`, with `h=0.0`.

2. **XY Model:** The XY model describes interactions between planar spins, or rotors. The energy is given by:

$$E(x) = -J \sum_{\langle i,j \rangle} \cos(\theta_i - \theta_j) \tag{44}$$

where $\theta_i = x_i$ represents the angle of the rotor at site $i$. The differences in angles between neighbors are computed using two distinct 2D convolutions for horizontal and vertical directions. We use a coupling constant of J=1.0.

For all targets, we consider both a deterministic (noise-free) setting and a stochastic setting where Gaussian noise is added to the scalar energy output during training. The noise standard deviation is automatically set to 10% of the empirical standard deviation of the energy function evaluated on a large batch of random inputs.

### E.2.2 ESTIMATOR ARCHITECTURES

We use two distinct neural network architectures for the gradient estimators, corresponding to whether the estimated field is constrained to be conservative.

1. **Conservative Estimator (for WGM-C, Sobolev, Surrogate):** This model, `Estimator_Potential_2D`, learns a scalar potential function, and its gradient is obtained via automatic differentiation. The architecture is a Convolutional Neural Network (CNN) consisting of three hidden `Conv2d` layers (16 channels each, 3x3 kernel) with `GroupNorm` and `GELU` activation functions. An adaptive average pooling layer and a final convolution map the features to a single scalar output.

2. **Non-Conservative Estimator (for WGM-NC, SPSA):** This model, `Estimator_Direct_2D`, directly outputs a gradient field of the same dimensions as the input. The architecture is a CNN with four `Conv2d` layers ($1 \rightarrow 16 \rightarrow 16 \rightarrow 16 \rightarrow 1$ channels, 3x3 kernel) using `GroupNorm` and `Tanh` activations.

### E.2.3 TRAINING AND EVALUATION PROTOCOL

- **Sampling Distribution:** Input samples $x$ are drawn from a standard multivariate normal distribution, $\mathcal{N}(0, I)$, on a 32x32 grid. Data is generated on-the-fly.

- **Training Parameters:** All models are trained for a maximum of 15 epochs using the Adam optimizer with a batch size of 1024. A grid search is performed over learning rates in the set {1, 0.5, 0.1, 0.05, 0.01, 0.005, 0.001, 0.0001}. For WGM-based methods, the learning rate is normalized by the number of input dimensions ($32 \times 32$).

- **Evaluation and Sampling Budget:** Performance is measured by the Mean Squared Error (MSE) and cosine similarity between the estimated and true gradients on a validation set. To ensure a fair comparison of function evaluations:
  - **WGM:** The number of Hutchinson vectors $M$ is varied in $\{1, 5, 10\}$. The total budget per sample is 1 evaluation of $J(x)$.
  - **Surrogate & Mixed:** The budget is 1 evaluation of $J(x)$ per sample.
  - **SPSA:** This method requires two evaluations ($J(x + \varepsilon v)$ and $J(x - \varepsilon v)$) per sample. To maintain an equivalent evaluation budget to other methods, SPSA is trained on mini-batches of half the size (512). The perturbation scale $\varepsilon$ is grid-searched over $\{1.0, 0.5, 0.1, 0.05, 0.001\}$.

### E.3 IMPLEMENTATION DETAILS FOR MNIST DENOISING GRADIENT EXPERIMENT

This section provides the implementation details for the gradient field reconstruction experiment on the MNIST dataset, as presented in Section 3.3.1 and Table 3 of the main paper. The goal is to learn the gradient of a Sum of Squared Errors (SSE) reconstruction cost under unknwon distribution.

### E.3.1 Target Function: SSE Reconstruction Error

The target function $J(x)$ measures the reconstruction error for a corrupted image $x$ with respect to a clean ground-truth image $x_{\text{clean}}$ from the MNIST dataset.

- **Corruption Process:** A clean image $x_{\text{clean}}$ is first corrupted by a downsampling-upsampling procedure. Specifically, we apply an average pooling operation with a $4 \times 4$ kernel, followed by bilinear upsampling back to the original $28 \times 28$ resolution. Finally, Gaussian noise $\epsilon \sim \mathcal{N}(0, 0.01^2 \cdot I)$ is added to produce the corrupted input $x$.

- **Objective Function:** The target scalar function is the Sum of Squared Errors (SSE):

$$J(x) = \|x - x_{\text{clean}}\|_2^2 = \sum_{i,j,c}(x_{i,j,c} - (x_{\text{clean}})_{i,j,c})^2 \tag{45}$$

  The analytical gradient of this function, which serves as the ground truth for evaluation, is $\nabla J(x) = 2(x - x_{\text{clean}})$. For training stability, the objective function is centered by subtracting its pre-calculated mean value, $J_c(x) = J(x) - \mathbb{E}[J(x)]$.

### E.3.2 Estimator Architectures

Two distinct CNN architectures are used to parameterize the gradient field estimators.

1. **Conservative Estimator (for Surrogate-C):** The `Estimator_Potential_2D` model learns a scalar potential $\phi_\theta(x)$, and the gradient is derived via automatic differentiation, $v_\theta(x) = \nabla\phi_\theta(x)$. The network consists of three `Conv2d` layers ($1 \to 16 \to 16 \to 1$ channels) with 3x3 kernels and `Softplus` activations. The final output is the mean over all spatial dimensions.

2. **Non-Conservative Estimator (for WGM-NC):** The `Estimator_Direct_GN_Tanh` model directly outputs a vector field of the same dimension as the input. The architecture is a three-layer CNN ($1 \to 32 \to 32 \to 1$ channels) with 3x3 kernels. Each of the first two convolutional layers is followed by `GroupNorm` and a `Tanh` activation function.

### E.3.3 Training and Evaluation Protocol

This experiment implements the "learning with the wrong score" strategy, where the sampling distribution's score is approximated using a Kernel Density Estimator (KDE) with a Gaussian kernel.

- **Sampling Distribution:** For each clean image $x_{\text{clean},i}$ in a batch, we generate a corresponding corrupted version $z_i$. The WGM estimator is then trained on samples $x$ drawn from a Gaussian distribution centered at $z_i$, i.e., $p(x|z_i) = \mathcal{N}(x|z_i, \sigma^2 I)$. The analytical score for this conditional distribution is $\nabla_x \log p(x|z_i) = -(x - z_i)/\sigma^2$. We use a fixed noise level of $\sigma = 0.3$.

- **Training Parameters:** All models are trained for 20 epochs using the Adam optimizer with a batch size of 2048 and a fixed learning rate of 0.005.

- **Evaluation:** Model performance is evaluated using the Mean Squared Error (MSE) and cosine similarity between the estimated gradient field and the true analytical gradient on the test set. We underlight that the distribution on the test set is the true MNIST distribution (not the noisy one). The model checkpoint with the lowest validation MSE is used for the final evaluation. The Hutchinson trace estimator for the divergence term in the WGM loss is computed using a single sample ($M = 1$).

## E.4 Other experiments

Neural network architecture and sampling distribution associated to the complementary experiment are reported on table 9. Computational ressources / training time are reported here:

| Experiment | NN Architecture | #Train Examples | Sampling Distribution |
|---|---|---|---|
| Toy 2D | MLP (4 layers, 256 units) | - | beta(1.01,1.01) |
| Lorenz 63 | ResNet | 30k | $\mathcal{N}(0,6)$ |
| N2D (Regression) | 5-layer CNN (tanh) | 5k | KDE with $\mathcal{N}(0,1)$ |
| N2D (VAE) | 5-layer CNN (tanh) | 5k | KDE with $\mathcal{N}(0,1)$ |

**Table 9:** Training configurations for the different experiments conducted in the paper. The parameters include the neural network architecture, the number of training examples, the sampling distribution used, and the number of gradient iterations learned.

| Experiment | Training Time | GPU Used |
|---|---|---|
| Toy 2D | 15 min | RTX 3090 |
| Lorenz 63 | 12 hours | RTX 3090 |
| Navier-Stokes 2D | 24 hours | 2xA100 |

**Table 10:** Training time and hardware specifications for each experiment.

