# OpenReview forum: "Data-driven estimation of gradient fields in the weak sense"
_ICLR.cc/2026/Conference — Submitted to ICLR 2026_

### Official Review · Reviewer_XyJu · 2025-10-24

**Soundness:** 4
**Presentation:** 2
**Contribution:** 3
**Rating:** 2
**Confidence:** 3

**Summary:**

The paper introduces a method for estimating gradient fields of functions whose gradient functional cannot be directly accessed or easily computed. The key insight is that a trick similar to score matching enables rewriting the loss to learn such functionals without requiring access to the target gradient. The authors empirically compare their method to existing benchmarks, demonstrating advantages in most cases.

**Strengths:**

The paper is well motivated. The main technique relies on an elegant connection to score matching, and its application to this setting is compelling. The proposed method is sensible and shows promise for tackling the problems identified by the authors.

**Weaknesses:**

There are several important areas where the authors could strengthen the paper. I am confident these can be addressed, but they require substantial changes that prevent acceptance at this venue.

- **Insufficient empirical verification.** The paper would benefit from additional ablation studies to understand how different parameters affect performance—for example, the number of Hutchinson vectors, the control variate $c$, and the choice of architecture for parametrizing $v_\theta$. Additionally, there is a notable gap between the practical problems where such a gradient field estimator would be valuable and the current shallow application of the proposed WGM method. Including one or two experiments in practically relevant settings would substantially strengthen the paper.
- **Structural issues.** While generally well written, the paper sometimes dwells on technical details and lacks clear motivation. For instance, Section 2.1 could be moved to the appendix, as most of its content is not essential to the main text. Section 2.3 needs better integration with the surrounding material. Section 4, in its current form, reads as a collection of equations that are difficult to parse without prior familiarity. The paper would also benefit from intuition about the control variate choice. Finally, Sections 4 and 5 lack proper motivation and clear connections to the rest of the paper.

I want to emphasize that I believe this will be a strong paper once these issues are addressed. I encourage the authors to incorporate this feedback and resubmit their work to a future conference.

**Questions:**

-

---

> ### Author Response · Authors · 2025-11-24
>
> We thank the reviewer for the candid feedback and for recognizing the potential of our method. We have worked intensively to address the "insufficient empirical verification" and "structural issues" you identified.
>
> **1. Insufficient Empirical Verification (Ablations)**
> > *"The paper would benefit from additional ablation studies [...] number of Hutchinson vectors, the control variate c"*
>
> We have added a comprehensive ablation study in the new **Appendix C**, as requested on the MNIST experiment where we target gradient field of a reconstruction cost:
> *   **Hutchinson Vectors ($M$):** **Figure 6 (Appendix C.1)** shows that the method is remarkably robust even with $M=1$ (single projection). Increasing $M$ reduces variance slightly but increases computational cost linearly; $M=1$ is sufficient for training due to the averaging effect of SGD.
> *   **Control Variate ($c$):** **Figure 7 (Appendix C.2)** show that choosing $c \approx \mathbb{E}[J]$ minimizes variance. Neglecting this ($c=0$) leads to instability.
>
> **2. Gap to Practice & New Experiments**
> > *"Including one or two experiments in practically relevant settings would substantially strengthen the paper."*
>
> We addressed this by adding two major experimental results:
> *   **Offline Black-Box Adversarial Attacks (Appendix D):** This is a highly relevant ML application. We show WGM can learn the gradient field of a classifier to generate adversarial examples for *new data* offline. For the same neural architecture and hyperparams tuning the WGM approaches seems to be competitive / outperform surrogate SPSA approaches.
> *   **CIFAR-10 Super-Resolution (Table 3):** We extended our results to natural images to demonstrate scalability beyond simple datasets even with the KDE trick.
>
> **3. Structural Issues**
> > *"Section 2.1 could be moved... Sections 4 and 5 lack proper motivation."*
>
> We followed your recommendations to improve the flow:
> *   **Reduced:** Section 2.1 (Sobolev norms)
> *   **Merged:** We integrated the "Inference" and "Mimicking" sections into the main **Method** section (now 3.4 & 3.5) to clarify the motivation *before* presenting the numerical results.
> *   **Refined:** The paragraph on Data assimilation is placed between the NS experiment to make the paper beginning of the paper easier to follow.
>
> **Continuous Improvement**
> We will continue to polish the manuscript and refine the bibliography throughout the discussion period. We hope that these substantial additions convince you that the paper is now ready for publication.
>
> We thank you again for your constructive criticism which has significantly improved the paper.

---

### Official Review · Reviewer_Jecu · 2025-10-28

**Soundness:** 1
**Presentation:** 2
**Contribution:** 2
**Rating:** 4
**Confidence:** 3

**Summary:**

This paper introduces WGM that estimates the gradient of a scalar field without requiring explicit differentiability. The method is inspired from score matching and extend to gradient of general scalar fields, with requirement that gradient of log likehood being known. Experiment results show superior performance of the proposed method compared to prior arts including finite difference or autodiff on scalar field surrogate models. Detailed derivation of the WGM loss is also provided.

**Strengths:**

- (Originality) This work proposes WGM as an indirect method for accurate estimation of gradient fields. The application of divergence theorem on top of score matching is novel and not studied in other works to the best of the reviewer's knowledge.

**Weaknesses:**

- (Significance and quality) The proposed WGM loss is tested only on two problems: Learning the gradient of MSE reconstruction error on MNIST dataset, and searching for the initial conditions of 2D NS equations with a non-differentiable physics solver. For the first problem, the practical meaning of constructing the GMM and estimating the gradient field of MSE reconstruction error is unclear. Basically, the proposed method seems to fit only for a very narrow class of problems and lacks general applicability (significance). It is very uncommon to have a need to predict the gradient field rather than the scalar field itself, while not having access to the gradient ground truth, is very uncommon.
- (Clarity) The paper is rather hard to follow as the structure doesn't follow the typical flow of deep learning research paper: Intro, Background, Method, Results, etc. The results section discusses a lot of method content, with actual experimental results shown in both section 3 and 5. Also, more experimental results should be included to support the claim.

**Questions:**

- The experiments discussed in this paper are more of domain specific problems with narrow significance. Even for pure comparison purpose, two case studies is not sufficient to prove the effectiveness of the proposed method. Is it possible to include at least one additional case, ideally with higher significance for comparison?
- There were papers using weak formulation in loss function such as ***wPINNs: Weak physics informed neural networks for approximating entropy solutions of hyperbolic conservation laws***, ***Weak-formulated physics-informed modeling and optimization for heterogeneous digital materials***, and ***The deep Ritz method: a deep learning-based numerical algorithm for solving variational problems***. Can you elaborate more on the fundamental difference between your approach and theirs?
- The framework requires prior knowledge of score values. This can be a major limitation in many problems. How can this be resolved generally? Also it's unclear how this is resolved in the 2D NS case study.

---

> ### Author Response · Authors · 2025-11-24
>
> We thank the reviewer for their detailed feedback. We have taken your concerns regarding the significance, the comparison with weak formulations in physics, and the paper structure very seriously. We have extensively revised the manuscript to address them.
>
> **Concerning your concerns about the significance of this work:**
>
> A Novel Fundamental Approach:While integration-by-parts estimators (Stein's method, Score Matching) have a rich history, to our knowledge, this is the first time this framework is generalized to estimate the gradient field of an arbitrary black-box function offline.
>
> Optimization Literature: The entier field of Gradient-Free Optimization exists precisely because obtaining gradients is difficult. Methods like CMA-ES (Hansen, 2016), Natural Evolution Strategies (Wierstra et al., 2014), or Zero-Order Optimization (Nesterov & Spokoiny, 2017).
>
>
> In RL, gradients of the environment are generally inaccessible,
>
> In Ocean/atmosphere, researchers infer surface currents from satellites observations of sea surface height (SSH) and then derivates it (surrogate approach) as SSC are (almost) never observed directly.
>
> Solving a Fundamental Dilemma: Until now, practitioners had only two flawed options: (1) Surrogate Modeling (learn $J_{\theta} \approx J$ then differentiate), which does not guarantee accurate estimate of the gradient as it is an indirect mean to achieve that goal (2) Finite Differences (SPSA), which suffer from high variance. WGM now offers a third path.
>
> Justification of Experimental Suite
>
> ***Experimental Scope & Updates***
>
> We fully understand your concern regarding the limited scope of the initial experiments. To address this, we have significantly expanded the empirical evaluation in the revised manuscript:
>
> **experimental Validation:** We moved the Gaussian Kernel / RBF experiment from the Appendix to the main text (Section 4.1.2) to provide clear intuition on linear parameterizations : as the learning problem is solved analitically this excludes optimization issue relative to NN training of the analysis.
>
> **Scalability:** We added a Super-Resolution experiment on CIFAR-10 (Table 3) to demonstrate scaling to natural images.
>
> **Practical Relevance:** As detailed above, we added the Offline Adversarial Attack experiment (Appendix D). This is a prototypical case where one needs to solve a learning problem (generating perturbations) against a black-box system, proving WGM's utility outside of physics.
>
> **Clarity:**
> We fully understand your concerns. We have extensively restructured the manuscript following the "Standard Flow" you suggested.
>
> ***Questions***
> Concerning your questions on PINNs:
> **Q2** "Can you elaborate more on the fundamental difference between your approach and theirs?"
>
> Your first reference seems relevant concerning the use of weak formulations :
> We clarified this fundamental distinction in Section 2.1. The difference lies in the knowledge of the operator:
> wPINNs: Solve a PDE where the operator is explicitly known (e.g., conservation laws). They employ the weak formulation specifically to handle low-regularity solutions (e.g., shocks) where strong derivatives are undefined.
> WGM (Ours): Addresses black-box function $J(x)$ where the gradient is unknown or inaccessible. We leverage integration by parts against the sampling distribution $p(x)$ to estimate the gradient of this unknown function without performing direct differentiation.
>
> A good example is: think about a really unregular object (that look like a brownian motion), even if you had access to derivatives they would tipically oscilaltes between very high/low values while weak formulations as ours only relies on functions values which in this 1d case would seems easier to handle.
>
> **Q3:** The framework requires prior knowledge of score values. How can this be resolved generally?
>
> We address this limitation in three ways:
>
>  KDE Approximation: We added experiments on MNIST and CIFAR-10 to demonstrate that a simple KDE estimate of the data is sufficient to apply WGM effectively.
>
> Navier-Stokes Case: In the 2D Navier-Stokes experiment, we apply the learning scheme described in Section 3.5 (Mimicking Gradient Descent). Here, the sampling distribution is defined as the marginal of the trajectories generated by the previous model iteration. This can be interpreted as a dynamic KDE approximation along the learned gradient flow, where the "score" is implicitly determined by the exploration noise we inject.
>
>  Generative Models: More generally, WGM can be applied with a trained generative model (e.g., Normalizing Flow, EBM, or Diffusion Model) on the data. One can then sample from this model and use its score $\nabla log p_\theta$. The problem then shifts from a score estimation problem to a standard generalization problem.
>
> We thank you again for your detailed review and hope that these clarifications and additional experiments address your concerns regarding the significance and applicability of our work.

---

> > ### Comment · Reviewer_Jecu · 2025-11-25
> >
> > Thank you for the detailed response.
> >
> > - The significance of the work is clear to me now. The main concern now is that the experiment results don't fully support the claim. In practice, the purpose of estimating gradient field is to perform optimization tasks. In the paper, however, most experiments only compare the estimated gradient to the true gradient on synthetic data. The author included an adversarial attack experiment, but there should be many recent techniques in addressing such task rather than gradient descent. Basically, most experiments lack practical meaning and seem to be synthesized just to show the effectiveness of WGM. Only the NS 2D experiment includes some optimization results that seem to have certain practical meaning but the setup isn't explained clearly. That said, **it is important to see a practical optimization task to be solved by WGM, with a comparison to SPSA and Surr in terms of optimal solution performance and computation effort**.
> > - The author mentioned the RL scenario and surface current inference for ocean and atmosphere. Would it be possible to include an experiment on these problems where the gradient field is actually meaningful and needed in the downstream task?
> > - Regarding potential approaches for obtaining score values, would it introduce additional computation effort? Also I can't find the explanation about the marginal of the trajectories for score estimation on the NS experiment in the manuscript.
> >
> > I'm open to adjust the rating if the above concerns are addressed, especially the one on experiment.

---

> > > ### Author Response · Authors · 2025-11-25
> > > **Important distinction between optimization and amortized inference**
> > >
> > > Thank you for your quick response!
> > >
> > > **Edit** Regarding to your comment : "the purpose of estimating gradient field is to perform optimization tasks" we add a Regarding your first point that "in practice, the purpose of estimating gradient field is to perform optimization tasks", we realized that the distinction between our setting and standard optimization might not have been clear enough. We have updated the Appendix to explicitly discuss Optimization vs. Amortized Inference.
> > >
> > > **1. Why Direct Optimization is Impossible in Real-Time 4DVar**:  In the Navier-Stokes data assimilation task, the cost function measures the discrepancy between the forecast of an initial condition and future observations (unseen at t0).
> > > Real-world constraint: At inference time (deployment), we only have access to past data. We cannot run an optimization loop (like SPSA or Gradient Descent) to minimize J because we cannot evaluate without the future data.
> > >
> > > Our Solution (Amortized Inference): We must train a model offline (using historical databases where the future is known) to predict the gradient update based solely on past data. WGM is used to train this predictor.
> > >
> > > **2. The Role of SPSA in our Experiments**
> > > Consequently, in our experiments, SPSA is not used as an online optimizer ((which would violate causality by peeking at the future). Instead, it is used as a baseline training objective: we train a neural network to regress the gradients estimated by SPSA. Our results demonstrate that WGM provides a much more robust training signal than finite differences for learning this conditional gradient field in high dimensions. Furthermore, learning the full gradient field theoretically enables conditional generative modeling via Langevin dynamics[1], allowing for diverse perturbations of the initial state for ensemble forecasting—a capability a simple point-optimizer does not provide.
> > >
> > > **3. Scalar vs. Gradient Fidelity (Geostrophic Currents)**
> > > To further illustrate the trade-off between a good scalar surrogate and a good gradient model, we will considering adding a discussion on the inverse problem associated to sea surface current estimation to exhibit the trade off between providing a good estimate of $J$ (the sea surface height) and $\nabla \times J$ ($\sim$ sea surface current) given constraints due to the parameterization of the model.
> > >
> > > Concerning the experiment on black box attack:
> > > We emphasize that our primary goal is to introduce a novel framework for gradient estimation in black-box settings (high dimensions, non-differentiable physics), rather than to claim state-of-the-art performance on established, highly-specialized benchmarks like adversarial attacks where domain-specific heuristics. This section serves an illustrative purpose, to address your first concerns concerning the utility of the proposed framework demonstrating the versatility of the WGM framework across radically different domains.
> > > [1]Implicit Generation and Modeling with Energy-Based Models

---

> > > > ### Author Response · Authors · 2025-11-25
> > > > **Concerning score estimation**
> > > >
> > > > We apologize for omitting the answer to your final question regarding the score estimation approaches and their computational costs.
> > > >
> > > > 1. Approaches for obtaining score values
> > > > The choice of method depends on the available data and resources. Fortunately, estimating the score of the data distribution is often more straightforward than training a full generative model for sampling. Common approaches include:
> > > >
> > > >     Denoising Autoencoders [1]: Learning to denoise inputs provides a direct estimator of the score.
> > > >
> > > >     Score Matching [2]: Directly training a model to match the data score.
> > > >
> > > >  Pre-trained Generative Models: If a pre-trained model (e.g., Normalizing Flow, EBM) is available, we can sample from it and use its exact score. In this case, the task becomes a generalization problem where WGM learns from samples associated with this model.
> > > >
> > > > 2. Computational Effort
> > > > In all cases, once the model providing the score is available, it is treated as a fixed function during the WGM training process. Its gradient is detached from the computational graph, meaning it does not require backpropagation. Therefore, there is no additional computational effort beyond the forward pass to evaluate the score, which is generally negligible compared to the cost of evaluating the complex black-box function.
> > > >
> > > > 3. Robustness to Approximation
> > > > This is why we believe our experimental results using a simple KDE proxy (Section 4.2) are particularly significant. They demonstrate that WGM remains robust and effective even when the score is approximated using a simple analytical form KDE, avoiding the need for heavy generative modeling in many practical scenarios.
> > > >
> > > > References:
> > > > [1] Vincent, P. "A Connection Between Score Matching and Denoising Autoencoders." (2011).
> > > > [2] Hyvärinen, A. "Estimation of Non-Normalized Statistical Models by Score Matching." (2005).

---

> ### Comment · Reviewer_Jecu · 2025-11-27
>
> I'm fine with the score estimation part and definition of SPSA baselines. But my biggest concern seems unaddressed yet: what do you need the gradient field for? What's the downstream task? From my experience, optimization is the most common downstream use for gradient field, or it can be something else. I fully agree that your approach can better estimate the gradient field better than finite difference or surrogate models, but in these black box problems, it gradient field really needed? There might be other direct methods developed, for instance the adversarial attack. Basically I'm wondering the **performance of your method on the downstream task (compared to other baseline methods that direct solve the downstream task) rather than the intermediate gradient estimation task.** This would reflect the practical meaning of your method. Inverse problem on sea current estimation is for sure a good example, but comparison with other inverse problem method is important.
>
> It's also fine to do the comparison on an optimization task where gradient descent is the best practice. In such case, the final optimized outcome is more important than the gradient field of a single step.

---

### Official Review · Reviewer_CcVK · 2025-10-28

**Soundness:** 3
**Presentation:** 3
**Contribution:** 2
**Rating:** 8
**Confidence:** 4

**Summary:**

The paper proposed a novel framework for estimation of gradient fields. The method is inspired by score matching and the goal is to learn $$\min_{\theta}\mathbb{E}_p \Vert  v(x) - \nabla J(x)  \Vert ^ 2$$

By integrate-by-part technique, the above problem is converted into learning the gradient fields in weak sense. The authors called the method, weak gradient matching (WGM). One technical detail is to use Hutchinson estimation of the divergence to estimate $\nabla \cdot v_{\theta}(x)$, a term emerging from integrate-by-part (see equation (5)).

Experiments include Gibbs energies, random CNNs, MNIST, Navier-Stokes. Results show that the proposed method has certain potential in various scenarios.

**Strengths:**

**Originality** The work is well-established from a novel perspective in learning gradient fields. The motivation is well presented. The method is established in a skillful way, e.g., the application of Hutchinson Estimation of divergence.

**Quality** The paper is presented in a way that methods are well referenced and related, e.g., Poincaré inequalities, Sobolev training and the works from score matching and continuous normalization of flows.

**Clarity**  The flow of the paper is smooth. From motivation to implementation, a novel framework is established.

**Weaknesses:**

1. My main concern is regarding experiments. It seems that the advantage of WGM is not apparent. See Table 2 and 3.
2. The inference procedure (Algorithm 1) may not be optimal. Instead one may seek $y$ such that $v_{\theta}(y)=\mathbf{0}$.

Typo:
1. Fig. 1, Left, right -> Top, bottom
2. Fig. 7, row 3,4,5 -> column 3,4,5?

**Questions:**

N.A.

---

> ### Author Response · Authors · 2025-11-24
>
> We thank the reviewer for the strong score and for recognizing the originality and skillful derivation of the proposed framework. We appreciate your feedback and have addressed your comments in the revised manuscript.
>
>    "My main concern is regarding experiments. It seems that the advantage of WGM is not apparent."
>
> We agree that in ideal low-noise regimes, WGM performs similarly to baselines. However, we have added new results to highlight its specific advantages in more challenging settings: High-Dimensional Scaling (CIFAR-10): We added a Super-Resolution experiment on CIFAR-10 (Table 3). Even when using an approximate score (KDE), WGM outperforms SPSA and Surrogate methods in terms of gradient cosine similarity on natural images.
>  Noise Robustness: Finite-difference methods (SPSA) are highly sensitive to noise in the target function J(x) : This require to properly tune the finite difference step (which we have done in our experiments ). Also one of the drawback of SPSA is that we need to have control on the experiment: it is not possible to traingradient field estimator solely from couples data, value $(x_i,J(x_i))$
>
> 2. Inference Procedure
>
>     "The inference procedure may not be optimal. Instead one may seek $ x^* $
>     such that $ \nabla v_{\theta}(x^*) = 0 $.
>
> This is indeed a valid alternative for finding stationary points and may constitute an interesting perspective (for example we could estimate the gradient field and use a second network to find these points).
> We replaced the title of the section "implicit regression" by "Amortized Variational Inference" (Section 3.4) is more accurate than "implicit regression" for our framework.
>
> 3. Manuscript Updates
> We have also corrected the captions for Figure 3 and Figure 7 as pointed out. Finally, following suggestions from other reviewers, we have restructured the paper (merging the methodological sections) to improve clarity, we decided to remove the part on VAE of the appendix to focus on less contributions but with better qualitative/quantitatives metrics.
>
> We thank you again for your detailed review and your strong support for the paper.

---

### Official Review · Reviewer_h9eY · 2025-11-01

**Soundness:** 3
**Presentation:** 2
**Contribution:** 3
**Rating:** 6
**Confidence:** 3

**Summary:**

This paper proposed an algorithm to estimate the gradient flow of a function which is not directly tractable (computational cost, blackbox etc).
It is based on the Green's identity and Hutchinson Estimation, yielding a weak gradient estimate that needs an explicit knowledge of the log probability. This assumption can be effectively weakened empirically by being replaced with a mixture of Gaussians.
The paper then carefully review the statistical estimates and convergence of the method.

**Strengths:**

Clear, sound and principled mathematical presentation of the subject.
Each hypothesis is carefully assessed and studied.

**Weaknesses:**

The overall structure is somewhat difficult to follow (different problems are presented one after the other, without it being clear exactly how they are related to each other).
The same applies to the presentation of the algorithm: it is somewhat difficult to follow what relates to the presentation of the algorithm, its adaptation to a particular case, etc.
The choice of experiments also seems somewhat arbitrary (at least it is not very clear whether these are classic problems or examples chosen by the author. In any case, a more in-depth discussion would not hurt).
Finally, I think there should be a more in-depth discussion with PINNs on the one hand, but above all with CMA-ES type methods, the purpose of which is precisely to estimate a gradient statistically.

**Questions:**

- In figure 2 top and bottom seems inverted (at least in caption).
- In section B.2 p17 (but this applies more generally) : use \big,\Big,\bigg,\Bigg for parenthesis, otherwise it is hard to identify each block between parenthesis.
- In B.3.2 p. 22 after polynomial regression, you have a reference missing.
- In algorithm 2 p 27 you have a typo : v or \upsilon is missing after Integrate.

---

> ### Author Response · Authors · 2025-11-24
> **Subject: Response to Reviewer h9eY: Structural improvements and comparisons with ES/PINNs**
>
> We thank the reviewer for the thorough reading and the positive assessment of the mathematical soundness of our work. We found your feedback on the manuscript's structure and the positioning relative to other methods extremely valuable. We have revised the paper accordingly.
>
> 1. Structure and Flow
> We agreed that the original organization was disjointed. We adopted a more classical structure for the revised manuscript ("Intro, Background, Method, Experiments") as suggested. Specifically, we moved technical background to the Appendix and integrated the inference schemes directly into the Method section to create a seamless narrative.
>
> 2. Comparison with CMA-ES and PINNs
> We thank the reviewer for this insightful suggestion. As Reviewer Jecu also requested a discussion on weak-form PINNs, we have incorporated a specific comparison in the revised manuscript (Section 2.1).
>
>     Weak-form PINNs: As noted, weak formulations have indeed been investigated in the context of PINNs. However, we clarify that wPINNs leverage integration by parts against a known PDE operator, whereas our work applies this principle to estimate unknown gradient fields from black-box samples.
>
>     Evolution Strategies (CMA-ES): As noted in Section 2.2, there is a fundamental distinction. CMA-ES is an online optimizer: it estimates a gradient direction locally to find a single optimum $x^*$
> and discards the information afterwards. In contrast, WGM is an offline estimator: it learns a functional approximation of the whole gradient flow $v_\theta = \nabla J$ over the entire data distribution. Once trained, WGM allows for reusable inference (e.g., sampling, attacks) without re-querying the black box.
>
> 3. Choice of Experiments
>
>     "The choice of experiments also seems somewhat arbitrary"
>
> We acknowledge that the selection might seem eclectic. The primary reason is the absence of established benchmarks specifically designed for offline black-box gradient field estimation (as opposed to standard black-box optimization benchmarks). Consequently, we constructed a progressive experimental suite to cover different regimes:
>
>
> -Classical Regression (Section 4.1.2): Validating theoretical properties on analytical functions.
>
>  -High-Dimensional Neural Estimators (Section 4.1.3 & 4.2): Testing scalability with Random CNNs when score is unknown (we added new experiment on targeting gradient field of a classifier on mnist and l2 reconstruction cost on CIFAR).
>
> -Concrete Scientific Application (Section 4.3): A realistic Navier-Stokes 4DVar problem where standard methods fail.
>
>
> Continuous Improvement
> Finally, we remain committed to polishing the manuscript throughout the discussion period. We will continue to refine the bibliography and fine-tune the positioning of our work relative to the literature to ensure the highest quality for the final version.
>
> Thank you again for your feedback, we remain at your disposals for complementary questions or improvement suggestions.

---

> > ### Author Response · Authors · 2025-11-25
> > **Edit: on the pratical difference between positioning of gradient free optimization methods & the proposed framework**
> >
> > **Edit:** In addition to the modifications to the main text, we have added an appendix section dedicated to clarifying the positioning between the proposed method and CMA-ES type methods on the Navier Stokes problem to explain why these methods are inapplicables in this setting.

---

### Author Response · Authors · 2025-11-24

Dear Area Chair and Reviewers,

We have uploaded the revised manuscript. As detailed in our previous responses, this version incorporates:
 **A complete structural overhaul** to align with standard deep learning paper presentation.

**New experiments:** CIFAR-10 reconstruction cost, and MNIST classification as new gradient targets, leveraging this new experiment we show in appendix D how to generate adversarial perturbations using the learned gradient field (offline attack).

We also decided to remove the part of the appendix dedicated to VAE in order to focus on less contributions with bette quantitative validation.

 **Comprehensive ablation studies (Appendix C).**

To ensure the full reproducibility of these new results, we will update the anonymous code repository to include the scripts and configurations corresponding to these additional experiments.

We remain available for any further questions during the discussion period.

---

### Author Response · Authors · 2025-12-03
**Overview of Major Revisions: New Experiments & Structural Updates**

Dear Area Chair,

We welcome you to this submission. Given the transition, we provide here a consolidated summary of the final manuscript to assist your assessment.

We explicitly thank the reviewers for their feedback. We have carefully incorporated all their suggestions to improve the manuscript. We were particularly encouraged by the constructive roadmap suggested by **Reviewer XyJu**, who recognized the **"excellent soundness"** of the method and noted that this work **"will be a strong paper once [specific] issues are addressed."**

We focused our revision on fulfilling these specific conditions (empirical verification & structure). Here is a summary of the major improvements:

**1. Strengthened Theoretical Foundation**
*Motivation:* To address questions regarding the necessity of direct gradient estimation versus standard surrogate modeling.
*   **New Analysis (Appendix A.4):** We added a theoretical derivation demonstrating why differentiating a surrogate model trained on scalar MSE is often mathematically suboptimal. We show that due to orthogonality issues in the approximation space, a good scalar fit does not guarantee an accurate gradient field. This provides the fundamental justification for our Weak Gradient Matching (WGM) framework.

**2. Extended Empirical Verification & Scalability**
*Motivation:* To address Reviewer Jecu's concern regarding the experimental scope and Reviewer XyJu's request for scalability.
*   **Gaussian Kernel Regression (Moved to Main Text, Section 4.1.2):** We moved this experiment from the Appendix to the main body to provide immediate analytical validation. This establishes a clear progression: from linear parameterizations to deep neural networks.
*   **CIFAR-10 Super-Resolution (Main Text, Section 3.3.1):** We integrated this new experiment into the core of the paper to demonstrate that WGM scales effectively to natural images using simple proxy scores (KDE).
*   **Sensitivity Analysis (Appendix C.1):** We added a rigorous ablation study (number of Hutchinson vectors, control variates), confirming the stability of the proposed hyperparameters.

**3. Demonstrated Downstream Utility**
*Motivation:* To prove the practical value of the estimated fields in concrete downstream tasks beyond intermediate metrics.
*   **Offline Black-Box Adversarial Attacks (Appendix C.3.2):** We demonstrate a machine learning application where WGM allows for *amortized* attack generation (offline training, real-time generation), offering a distinct advantage (and methodological positionning) over the litterature of gradient free online optimization methods (e.g. CMA-ES).
*   **Geostrophic Current Estimation (Appendix C.3):** A real-world oceanography inverse problem where the underlying object of interest is the gradient of a function and traditionally only accessible through the differentiation of a surrogate model.
*   **Clarification on Amortized Inference:** We clarified the Navier-Stokes 4DVar experiment to highlight that direct optimization is causally impossible at test time (missing future data), making our learned conditional gradient predictor essential.

**4. Structural Improvements**
The manuscript has been reorganized into a standard flow (Background $\rightarrow$ Method $\rightarrow$ Experiments), with technical preliminaries moved to the Appendix to improve readability.

We believe that by fulfilling the specific requests for empirical scaling and restructuring, this final manuscript now realizes the potential identified by the reviewers.

Best regards,
The Authors.

---

### Meta-Review · Area_Chair_cVVZ · 2026-01-06

**Summary:**

Reviewers acknowledged the originality and mathematical soundness of the proposed weak gradient matching framework, and the rebuttal substantially improved the manuscript through major structural reorganization, additional experiments, ablation studies, and more precise positioning with respect to related methods such as PINNs, CMA-ES, and score matching. These revisions addressed many concerns regarding clarity, empirical robustness, and methodological motivation. However, several reviewers remained unconvinced about the practical significance of estimating full gradient fields as an intermediate objective. In particular, concerns persist about the lack of compelling downstream task evaluations demonstrating that WGM provides clear advantages over methods that directly solve the target problems. As a result, despite being very close to the acceptance threshold, I recommend rejection, while encouraging the authors to revise and resubmit after strengthening the downstream task evaluation and clarifying the practical impact of the proposed approach.

**Reviewer Concerns:**

Reviewer h9eY:
Concerns regarding manuscript structure, clarity of the algorithmic presentation, and positioning relative to PINNs and CMA-ES were addressed mainly through a significant reorganization of the paper and an added comparative discussion. The choice of experiments was better motivated in the rebuttal. However, the practical impact of learning complete gradient fields, beyond illustrative examples, remains only partially convincing.

Reviewer CcVK: Additional experiments on CIFAR-10 and robustness analyses partially addressed concerns about the visibility of WGM’s empirical advantage. Clarifications on inference procedures and terminology improved readability. Nonetheless, the reviewer’s skepticism regarding whether WGM offers a decisive empirical benefit over strong baselines in realistic settings remains somewhat outstanding.

Reviewer Jecu: Several concerns about significance, relation to weak-form PINNs, score estimation, and manuscript structure were carefully addressed, and the novelty of the framework is now clearer. However, the central concern—whether estimating a gradient field is necessary or beneficial for meaningful downstream tasks, compared with methods that solve those tasks directly—remains unresolved and remains the main outstanding issue.

Reviewer XyJu: Concerns about insufficient ablations and structural issues were substantially addressed through new experiments, extensive ablation studies, and improved organization. Still, the gap between gradient-field estimation accuracy and clear, compelling downstream task performance was not fully closed, leaving doubts about the practical relevance of the proposed method.

**Reviewer Scores:**

Reviewer h9eY: The score would likely remain unchanged, as the rebuttal significantly improved structure and positioning but did not fundamentally alter the perceived impact or clarify a compelling downstream use case.

Reviewer CcVK: The score would remain unchanged, since the reviewer already viewed the contribution positively and the rebuttal mainly strengthened empirical support without changing their overall assessment.

Reviewer Jecu: The score would likely remain unchanged, because while several clarifications were provided, the core concern about downstream task relevance and end-to-end practical benefit was not fully resolved.

Reviewer XyJu: The score would likely remain unchanged, as the rebuttal addressed ablations and structure but did not sufficiently close the gap between gradient-field estimation and practically compelling downstream applications.

---

### Decision · Program_Chairs · 2026-01-26

Reject